# Complex-Edit: CoT-Like Instruction Generation for Complexity-Controllable Image Editing Benchmark

**Siwei Yang**                                                                    *syang217@ucsc.edu*
*Computer Science and Engineering Department, UC Santa Cruz*

**Mude Hui**                                                                       *muhui@ucsc.edu*
*Computer Science and Engineering Department, UC Santa Cruz*

**Bingchen Zhao**                                                        *bingchen.zhao@ed.ac.uk*
*School of Informatics, University of Edinburgh*

**Yuyin Zhou**                                                                   *yzhou284@ucsc.edu*
*Computer Science and Engineering Department, UC Santa Cruz*

**Nataniel Ruiz**                                                        *natanielruiz@google.com*
*DeepMind, Google*

**Cihang Xie**                                                                     *cixie@ucsc.edu*
*Computer Science and Engineering Department, UC Santa Cruz*

**Reviewed on OpenReview:** *https://openreview.net/forum?id=lL1JR6dxG8*

## Abstract

We introduce `Complex-Edit`, a comprehensive benchmark designed to systematically evaluate instruction-based image editing models across instructions of varying complexity. To develop this benchmark, we harness GPT-4o to automatically collect a diverse set of editing instructions at scale. Our approach follows a well-structured "Chain-of-Edit" pipeline: we first generate individual atomic editing tasks independently and then integrate them to form cohesive, complex instructions. Additionally, we introduce a suite of metrics to assess various aspects of editing performance, along with a VLM-based auto-evaluation pipeline that supports large-scale assessments. Our benchmark yields several notable insights: 1) Open-source models significantly underperform relative to proprietary, closed-source models, with the performance gap widening as instruction complexity increases; 2) Increased instructional complexity primarily impairs the models' ability to retain key elements from the input images; 3) Stronger models aren't necessarily more resilient towards higher complexity; 4) Decomposing a complex instruction into a sequence of atomic steps, executed in a step-by-step manner, substantially degrades performance across multiple metrics; 5) A straightforward Best-of-N selection strategy improves results for both direct editing and the step-by-step sequential approach; and 6) We observe a "curse of synthetic data": when synthetic data is involved in model training, the edited images from such models tend to appear increasingly synthetic as the complexity of the editing instructions rises — a phenomenon that intriguingly also manifests in the latest GPT-Image-1's outputs. The code for evaluation and data generation, and the test set is released at https://github.com/UCSC-VLAA/Complex-Edit.

## 1 Introduction

Since InstructPix2Pix (Brooks et al., 2023) introduced instruction-based image editing—where images are directly modified through textual commands—the field has witnessed remarkable progress. Noteworthy developments include enhancements in the quality of training images (Kawar et al., 2023; Zhang et al.,

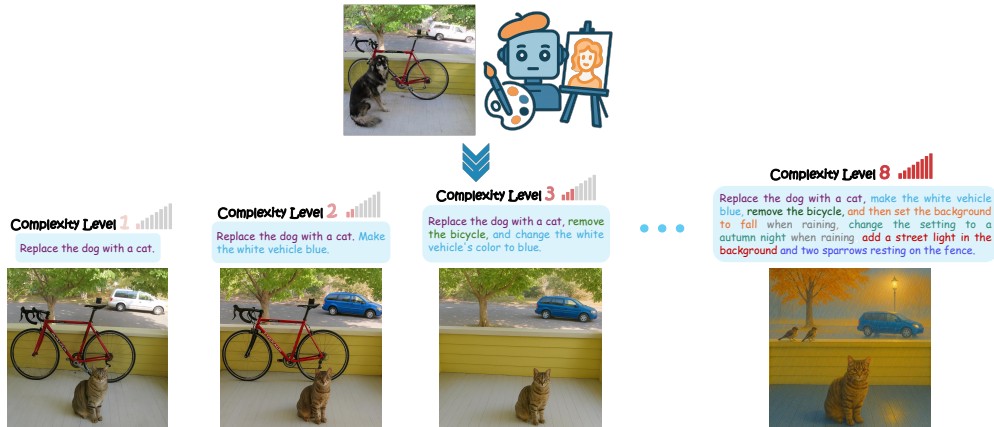

Figure 1: An illustration of our `Complex-Edit`. This figure presents a progression of instruction complexity in image editing tasks, highlighting the transition from atomic edits to highly intricate transformations.

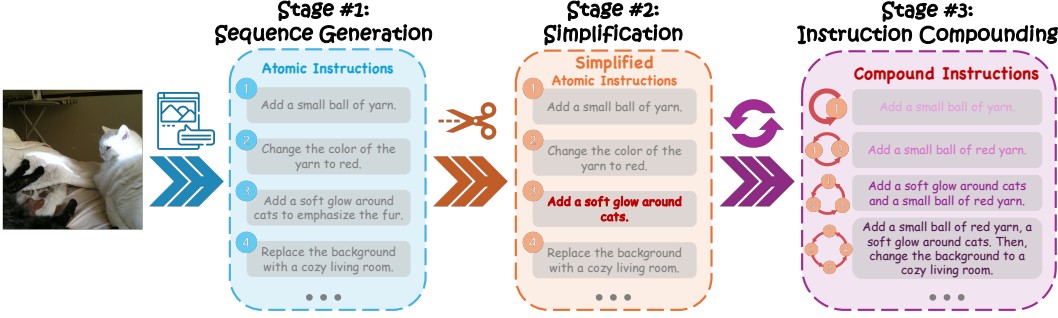

Figure 2: An overview of our data collection pipeline. The pipeline consists of three distinct stages: 1) **Stage #1 Sequence Generation**: for each image, a series of atomic instructions is produced; 2) **Stage #2 Simplification**: each fundamental instruction is refined to eliminate extraneous details, preserving only the essential description of the editing process; 3) **Stage #3 Instruction Compounding**: several atomic instructions are integrated into one comprehensive instruction.

2024; Hui et al., 2025; Zhao et al., 2024; Wei et al., 2025; Baldridge et al., 2024), the evolution of model architectures from CNN-based to transformer-based diffusion models (Ma et al., 2024), the refinement of training methodologies (Simsar et al., 2024; Shi et al., 2024), and the on-going exploration of test-time scaling (Guo et al., 2025; Ma et al., 2025).

Despite these strides, much of the available training data and evaluation benchmarks rely on relatively simple editing instructions (Brooks et al., 2023; Zhang et al., 2023; 2024; Zhao et al., 2024; Yu et al., 2024; Wei et al., 2025; Sheynin et al., 2023; Ku et al., 2024b). Yet, real-world scenarios often require handling instructions that vary considerably in complexity. For instance, while one user might simply request to "remove the car", another might need a more elaborate transformation, such as "replace the car with a blue bus featuring an iPhone advertisement on its side". The current lack of evaluation benchmarks capable of handling such diverse instruction complexities not only impedes our ability to rigorously assess existing editing models but also hampers the evaluation of emerging, more powerful systems (Guo et al., 2025; Ma et al., 2025).

To bridge this gap, we present a novel data generation pipeline that leverages advanced GPT models to create a diverse, scalable, and complexity-controllable evaluation dataset for image editing (see Fig. 2 for an illustration). Our approach follows a chain-of-thought-like (Wei et al., 2022) paradigm, unfolding in three key stages. First, in the *Sequence Generation* stage, an input image prompts GPT-4o to generate a series of simple instructions corresponding to predefined atomic operations (*e.g.*, "Add an Object", "Change the Object Color"), with each serving as an intermediate step toward a more complex editing task. Next, recognizing that these GPT-generated atomic instructions may contain superfluous/unnecessary details—such as added commentary on the operation's intention—we then move to a *Simplification* stage, where each

instruction is trimmed to retain only its core editing intent. In this final *Instruction Compounding* stage, the simplified atomic instructions are merged into a single, coherent complex instruction (again, via GPT-4o), using the input image as contextual guidance. We name this new dataset `Complex-Edit`. Importantly, this structured process of creating `Complex-Edit` allows us to quantitatively control the editing instruction's complexity — defined as the number of atomic operations in a single instruction — by simply adjusting the number of merged atomic instructions. This benchmark directly scales the **semantic complexity** of the task: as the number of atomic operations increases, the instruction inherently compels the model to process denser linguistic information and resolve increasingly intricate dependencies between visual elements. We further discuss the theoretical positioning of this compositional complexity within **a larger hierarchy of complexity types** for image editing in Appendix Sec. A.

To evaluate performance on these complex instructions, we developed a comprehensive evaluation framework that measures three critical dimensions: (1) *Instruction Following*, which assesses whether the intended modifications are correctly applied; (2) *Identity Preservation*, which ensures that unspecified elements remain unchanged; and (3) *Perceptual Quality*, which evaluates the overall aesthetic quality and absence of artifacts. To support large-scale evaluations, we implement a vision language model (VLM)-based autograding system to quantify these dimensions. Our investigation further identifies important considerations in enhancing this evaluation pipeline. For example, enabling chain-of-thought reasoning in VLM evaluations does not necessarily enhance evaluation quality—contrary to previous findings (Ku et al., 2024a; Hui et al., 2025)—while providing detailed rubrics and using direct numeric scoring (Xie et al., 2023; 2024; Zhou et al., 2024; Cui et al., 2024) consistently improves autograder performance.

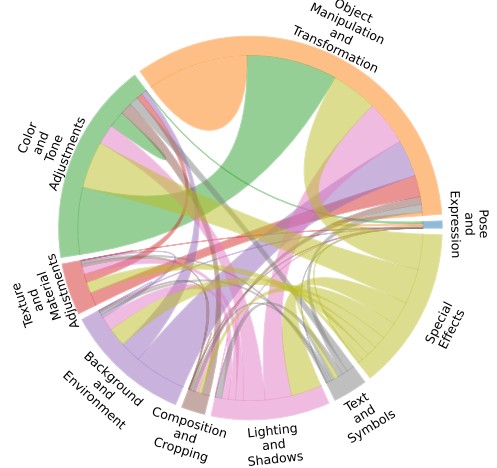

Figure 3: The distribution of atomic instructions among 9 operation categories. The arc thickness between two categories shows the number of adjacent instructions from these two categories.

Our extensive evaluation results reveal five key insights that were previously difficult to capture with existing benchmarks. First, open-source models significantly underperform compared to proprietary, closed-source models, with this disparity growing as editing complexity increases. Second, the increasing instruction complexity mainly affects editing models' identity preservation, while its impact on instruction following varies across models. Thirdly, more capable models do not reliably exhibit increased resilience to the negative effects of the higher instruction complexity. Fourth, unlike the benefits observed with chain-of-thought (CoT) reasoning in text generation, CoT-inspired sequential editing yields much worse results than directly executing complex instructions — we note this is because sequential editing significantly degrades outputs in instruction following, identity preservation, and perceptual quality. Fifth, simple techniques such as Best-of-N sampling improve direct editing outcomes across all three dimensions and, when applied to sequential editing, notably enhance identity preservation and perceptual quality. Lastly, our `Complex-Edit` reveals an emerging 'curse of synthetic data' — when training utilizes synthetic images, models tend to produce edited outputs with overly synthetic qualities under complex instructions, often resembling oil paintings or animations. More interestingly, this phenomenon is also observed with the latest GPT-Image-1, potentially suggesting that the incorporation of synthetic data may contribute to GPT-Image-1's advanced image generation capabilities.

We hope this newly developed benchmark, `Complex-Edit`, will not only deepen the understanding of instruction-based image editing models but also serve as a valuable framework for the rigorous evaluation of next-generation image editing systems, particularly those with test-time scalability.

## 2 Collection of `Complex-Edit`

As illustrated in Fig. 2, the collection of `Complex-Edit` is organized into three stages: 1) **Stage #1 Sequence Generation**: a sequence of atomic instructions are generated for each image; 2) **Stage #2 Simplification**:

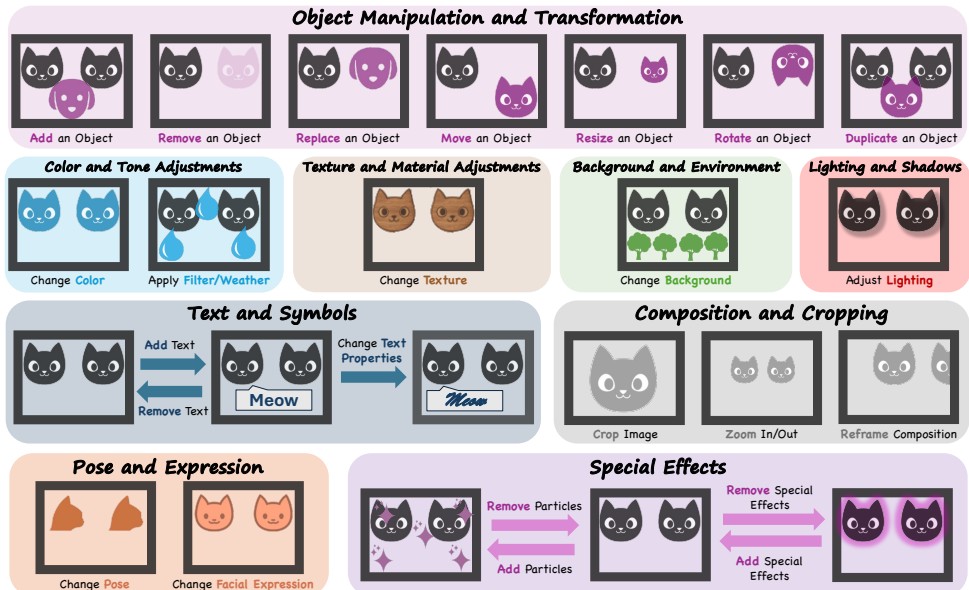

Figure 4: An illustration of 24 types of atomic editing operations in 9 categories.

each atomic instruction is simplified to remove unnecessary information other than the description of the editing operation; 3) **Stage #3 Instruction Compounding**: multiple atomic instructions are combined into a single, complex instruction.

## 2.1 Sequence Generation

In this first stage, we define 24 distinct atomic operations that represent the most basic actions of image editing, which are grouped into 9 categories — *Object Manipulation and Transformation*, *Color and Tone Adjustments*, *Texture and Material Adjustments*, *Background and Environment*, *Lighting and Shadows*, *Text and Symbols*, *Composition and Cropping*, *Pose and Expression*, *Special Effects* — each of which captures a unique aspect of image editing, as shown in Fig. 4.

We provide detailed descriptions of each operation type in the Sec. B. These categorized atomic operation types and their descriptions, along with an input image, are fed into GPT-4o to generate a sequence of atomic instructions with a pre-determined length. Each generated instruction is associated with one of the predefined atomic operations. We also record the atomic operation type for each instruction, with its distribution illustrated in Fig. 3. This metadata can facilitate the future development of specialized image editing models.

## 2.2 Simplification

In practice, these GPT-generated instructions sometimes are unnecessarily rich in detail and may include extraneous commentary on the intentions of the editing operations, as shown in Fig. 2. Because our benchmark is designed to evaluate the performance of image-editing models on clear and concise instructions, we employ a dedicated simplification stage to eliminate any superfluous content. Specifically, each generated atomic instruction is examined by GPT-4o to determine whether it requires further simplification; if so, GPT-4o outputs a simplified version adhering to the predefined response format, and the original instruction is replaced accordingly.

## 2.3 Instruction Compounding

Lastly, simplified atomic instructions are combined into compound instructions. Specifically, given a sequence of $N$ atomic instructions, we progressively generate $N$ compound instructions corresponding to different complexity levels: a compound instruction at complexity level $C_i$ is formed by combining the first $i$ atomic instructions in the sequence, with the simplest level ($C_1$) being identical to the first atomic instruction and the hardest level ($C_N$) integrating all $N$ atomic instructions.

In implementation, rather than simply concatenating the instructions, atomic instructions are seamlessly integrated, which allows potential reordering or merging to produce to make the final outcome more natural and coherent. For example, in Fig. 2, rather than sequentially executing "add a ball of yarn" and "change the color of the yarn to red", the instructions are compounded into a single directive: "add a red ball of yarn".

Table 1: Random sampling vs Discriminative sampling.

| Deterministic | Correlation | | |
|:---:|:---:|:---:|:---:|
| | *IF* | *IP* | *PQ* |
| | **0.468** | **0.530** | **0.234** |
| ✓ | 0.461 | 0.507 | 0.215 |

Table 2: Results from different scoring methods. We can observe that w/ COT and w/ Rubric are the best for IF and IP, and w/o COT and w/ rubric are the best for PQ.

| Scoring Method | CoT | Rubric | Correlation | | |
|:---|:---:|:---:|:---:|:---:|:---:|
| | | | *IF* | *IP* | *PQ* |
| Token Prob | | | 0.411 | 0.366 | 0.136 |
| Token Prob | ✓ | | 0.447 | 0.460 | 0.158 |
| Numeric | | | 0.434 | 0.450 | 0.207 |
| Numeric | | ✓ | 0.446 | 0.451 | **0.234** |
| Numeric | ✓ | ✓ | **0.468** | **0.530** | 0.208 |
| | | | CLIP$_{dir}$ | CLIP$_{img}$ | |
| | | | 0.182 | 0.523 | |

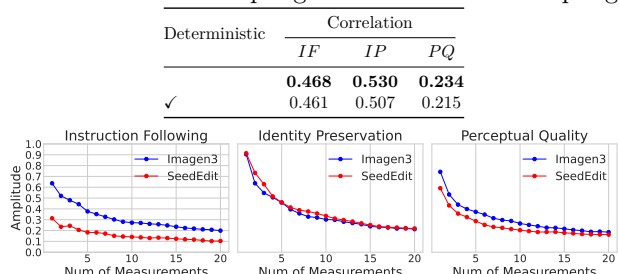

Figure 5: Relationship between per-sample variance

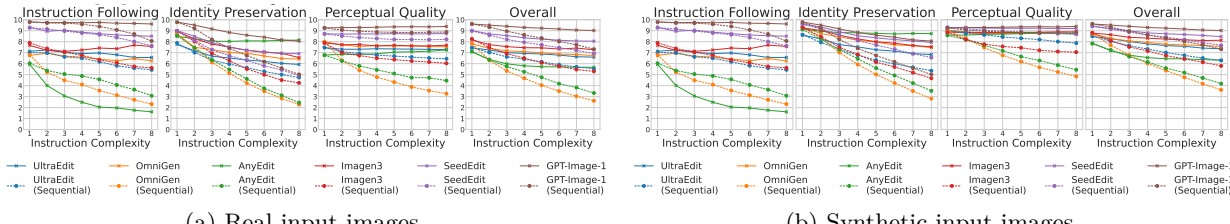

(a) Real input images.      (b) Synthetic input images.

Figure 6: Evaluation results of direct and sequential editing. The instruction complexity level ranges from $C_1$ to $C_8$. This indicates that $IP$ and $PQ$ consistently dropped as the instruction complexity level grows, while it influence on $IF$ varies among different models. Also, the performance gap between models is enhanced with increasing complexity.

## 2.4 Implementation Details

To ensure that GPT-4o fully understands our objectives, we carefully construct multiple few-shot examples for each of the three stages. Additionally, CoT reasoning (Wei et al., 2022) is enabled in both Stage #1 (Sequence Generation) and Stage #3 (Instruction Compounding) to enhance generation quality. Furthermore, to improve data diversity, we slightly increase the sampling temperature from 1.0 to 1.15 during Stage #1 (Sequence Generation). Although this adjustment promotes diversity, it also increases the likelihood of generating garbled text. To address this, we implement a rule-based filtration mechanism at each stage that detects and regenerates any flawed outputs.

## 3 Evaluation

Following prior works (Ku et al., 2024a; Hui et al., 2025), we employ VLM-based autograders to facilitate large-scale evaluation. However, we have observed that these frameworks do not fully capture the nuances of instruction-guided image editing. In response, we introduce several enhancements to address these limitations and improve the overall autograding framework.

## 3.1 Metric Design

Our evaluation framework focuses on two primary dimensions: 1) the alignment dimension—whether the output image reflects the changes specified by the editing instruction, and 2) the perceptual quality dimension— whether the output image looks aesthetically pleasing and is devoid of visual artifacts.

### 3.1.1 Alignment

Unlike HQ-Edit (Hui et al., 2025), which simply measures alignment as a single metric, we hereby decompose it into two complementary criteria: 1)**Instruction Following** ($IF$) measures whether the specified modifications are present in the output image; 2)**Identity Preservation** ($IP$) assesses whether elements of the input image that should remain unchanged are indeed preserved.

We would like to point out that these two criteria roughly correspond to the directional CLIP similarity (CLIP$_{dir}$) and image-wise CLIP similarity (CLIP$_{img}$) metrics used in earlier studies (Brooks et al., 2023; Zhang et al., 2023; 2024). When accessing $IF$ and $IP$, we feed the VLM with a triplet containing {*input*

*image*, *output image*, and *editing instruction*}. We provide a series of example results of *IF* and *IP* in Appendix Fig. 16.

### 3.1.2 Perceptual Quality

Beyond alignment, we also evaluate the overall visual quality of the generated images. Specifically, our **Perceptual Quality (PQ)** metric examines factors such as consistency in lighting and shadows, style coherence, and the seamless integration of elements. As users may sometimes request bizarre or non-aesthetic edits, such as "add motion blur to these vehicles" or "enlarge the pet to match the height of the owner", one might argue that it is essential to include editing instructions when evaluating perceptual quality. However, as discussed in Sec. 4.1.4, our empirical results indicate that providing editing instructions to VLMs actually reduces the correlation between VLM and human evaluations. Consequently, for *PQ* evaluation, we only supply the VLM autograder with the edited image. Sample *PQ* results are provided in Appendix Fig. 17.

### 3.1.3 Overall Score

Lastly, we define the overall score, $O$, simply as the arithmetic mean of the three metrics to summarize performance: $O = \frac{IF+IP+PQ}{3}$.

## 3.2 Metric Calculation

### 3.2.1 Numeric Scoring v.s. Token Probability

In line with VIEScore and HQ-Edit, we instruct the VLM to assign a score between 0 and 10 for each metric. We refer to this method as **numeric scoring**. In addition, we explore another approach for metric computation: **token probalility**, which is widely used in test-time scaling and VLM post-training (Xie et al., 2023; 2024; Zhou et al., 2024; Cui et al., 2024). Specifically, rather than directly asking the VLM for a numerical score, we reformulate the evaluation as a binary classification task by posing a *yes-or-no question* to the VLM, *e.g.*, "Do the specified changes appear in the output image?". The score is then calculated as the normalized token probability for the affirmative response "Yes" using $\text{Prob}_{\text{Yes}}/(\text{Prob}_{\text{Yes}} + \text{Prob}_{\text{No}})$.

### 3.2.2 Detailed Rubric

In contrast to existing frameworks like VIEScore and HQ-Edit, we design comprehensive rubrics for each metric to guide the evaluation process and improve interpretability. Detailed descriptions of these rubrics are provided in the Sec. C.1. We discuss the impact of these rubrics and different scoring approaches in Sec. 4.1.2.

## 3.3 Per-sample Variance

Previous evaluations of image-editing models using advanced proprietary VLMs, such as GPT-4o (Hurst et al., 2024) and Gemini (Team et al., 2023; 2024), have encountered issues with score stability, leading to score variations for individual samples across different runs. Although averaging over many samples generally mitigates these discrepancies at the dataset level, per-sample stability is crucial for test-time scaling, where accurate and consistent evaluation is required for each individual sample (Guo et al., 2025; Ma et al., 2025).

A straightforward approach to enhance stability utilizes multiple measurements to average the resultant scores, though this approach may considerably escalate computational costs. Alternatively, one can attempt to make VLM outputs more deterministic by using greedy random sampling techniques (*e.g.*, by setting the sampling probability mass to an extremely low value such as 1e-7). However, empirically, this deterministic approach slightly lower the correlation with human evaluations as shown in Tab. 1. Consequently, we adopted the first approach by default, utilizing the average score from 20 independent evaluations for each sample. Further discussion on reducing per-sample variance is included in Sec. 4.2.

Table 3: Evaluation results with direct editing on real images with the instruction complexity at $C_8$.

| Model | Prop. of Real Images | Photorealism | IF | IP | PQ | O |
|---|---|---|---|---|---|---|
| UltraEdit | ✓ | ★★★ | 6.56 | 5.93 | 7.29 | 6.59 |
| OmniGen | ✓✓✓ | ★★★★ | 6.25 | 6.42 | 7.54 | 6.74 |
| AnyEdit | ✓✓✓ | ★★★★★ | 1.60 | **8.15** | 7.25 | 5.67 |
| SeedEdit | ✓ | ★★★ | 8.49 | 6.91 | 8.74 | 8.04 |
| Imagen3 | ? | ★★★ | 7.56 | 6.55 | 7.67 | 7.26 |
| GPT-Image-1 | ? | ★ | **9.61** | 8.06 | **9.39** | **9.02** |

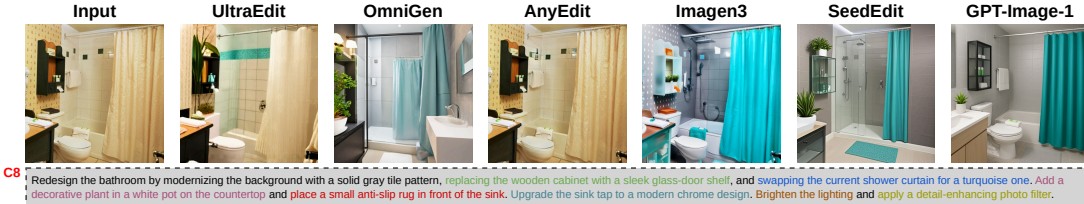

| **Input** | **UltraEdit** | **OmniGen** | **AnyEdit** | **Imagen3** | **SeedEdit** | **GPT-Image-1** |

**C8** Redesign the bathroom by modernizing the background with a solid gray tile pattern, replacing the wooden cabinet with a sleek glass-door shelf, and swapping the current shower curtain for a turquoise one. Add a decorative plant in a white pot on the countertop and place a small anti-slip rug in front of the sink. Upgrade the sink tap to a modern chrome design. Brighten the lighting and apply a detail-enhancing photo filter.

Figure 7: A real image edited with a $C_8$ instruction. Some models' output lose the photorealism completely.

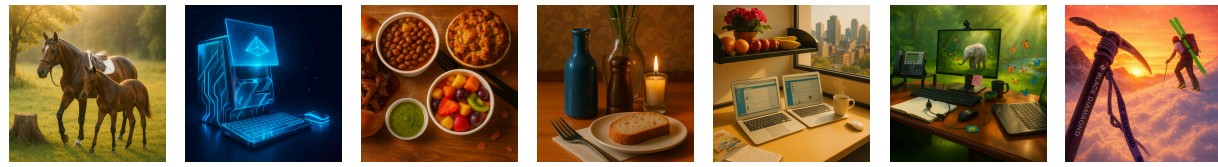

Figure 8: Real images edited with $C_8$ instructions by GPT-Image-1. Outputs from GPT-Image-1 severely lose the photorealism. See additional results in Appendix Fig. 23.

# 4 Meta-Evaluation for Metrics

In this section, we detail our meta-evaluation of the VLM-based auto-evaluation framework, focusing on design choices and their effectiveness in assessing image editing model outputs. We expect an effective auto-evaluation pipeline to satisfy two core criteria: 1) High correlation with human judgments, and 2) Low variance for ensuring reproducibility. Next, we first examine how various design decisions impact correlation with human evaluations (Sec. 4.1) and then describe our strategies for reducing per-sample variance (Sec. 4.2).

## 4.1 Correlation with Human Evaluation

### 4.1.1 Meta-Evaluation Settings

Meta-evaluation for instruction-based image editing metrics (Ku et al., 2024a; Hui et al., 2025) is typically performed by correlating human-assigned scores with metric values. However, we found that expecting human raters to assign consistent numeric scores across diverse samples is challenging. Instead, for each input image and editing instruction, we present raters with a pair of output images and ask them to compare the outputs with respect to each metric. We then compute the correlation between these human comparisons and the differences in the corresponding metric scores. Additionally, we ask raters to choose their preferred output based on overall impression. This comparative approach allows us to align our metric aggregation with human intuition and to determine the best way to combine the three proposed metrics into a single score.

To ensure that our meta-evaluations generalize across varying levels of instruction complexity, we sampled 100 output images for each complexity level from $C_1$ (*i.e.*, the simplest level, just a single atomic instruction) to $C_8$ (*i.e.*, the hardest level, compounding from 8 atomic instructions) using both Imagen3 (Baldridge et al., 2024) and SeedEdit (Shi et al., 2024), resulting in 800 image pairs. Each pair was annotated independently by at least two human raters. We use GPT-4o as our VLM evaluator for the meta-evaluation study. Since the per-sample variance is yet to be discussed in Sec. 4.2, for now, we *naively* evaluate each sample 40 times and average the scores to reduce the per-sample variance.

### 4.1.2 Scoring Method and Rubric

As shown in Tab. 2, our experiments indicate that numeric scoring yields a higher correlation with human evaluation than token probability methods. Moreover, adding a detailed rubric to numeric scoring enhances this correlation consistently, highlighting the need for clear evaluation guidelines.

### 4.1.3 Chain-of-Thought

We additionally investigate the impact of CoT on our evaluation by adding an explicit instruction (*e.g.*, "explain your reasoning before answering the question") to encourage the VLM evaluator to articulate its reasoning prior to delivering the final score. The complete prompt for evaluation is provided in Appendix Sec. C.2. However, as shown in Tab. 2, our results indicate that CoT fails to enhance metric correlation consistently. In particular, CoT negatively affects the correlation for Perceptual Quality $PQ$ when using numeric scoring. Consequently, we utilize numeric scoring with detailed rubrics and CoT only for Instruction Following $IF$ and Identity Preservation $IP$, while CoT is disabled during Perceptual Quality $PQ$ evaluation.

### 4.1.4 Instruction Input for Perceptual Quality

One might initially hypothesize that including the editing instructions alongside the output image could help the VLM discount aesthetically inferior modifications that are explicitly required by the instructions during Perceptual Quality $PQ$ evaluation. However, our experiments reveal that providing the editing instructions significantly lowers the correlation with human evaluations from 0.234 to 0.046. Consequently, we exclude the editing instructions when evaluating Perceptual Quality $PQ$, ensuring that the VLM's assessments of image quality better align with human perception.

### 4.1.5 Different Averaging Formula for Overall Score

Previous studies (Ku et al., 2024a; Wei et al., 2025) advocate using the geometric mean to combine individual metrics to penalize low scores. In our case, switching from the geometric mean to the arithmetic mean resulted in a modest correlation increase from 0.334 to 0.386. Therefore, we compute the overall score as the arithmetic mean of the individual metrics.

### 4.1.6 Comparison with CLIP Scores

We further compare the correlation of Directional CLIP Score $\text{CLIP}_{\text{dir}}$ and Image-wise CLIP Score $\text{CLIP}_{\text{img}}$ with human evaluation, as these scores similarly assess aspects of Instruction Following $IF$ and Identity Preservation $IP$. As reported in Tab. 2, these CLIP scores exhibit lower correlations with human evaluations compared to our metrics.

## 4.2 Per-Sample Variance

As discussed in Sec. 3.3, one way to reduce per-sample variance is to set the probability mass for sampling to an extremely small value, *e.g.*, 1e-7, thereby approximating the determinism of proprietary VLMs. However, as shown in Tab. 1, this approach slightly diminishes the correlation with human evaluations. Consequently, we disable deterministic sampling and instead perform multiple evaluations per sample, using the average score as the final metric.

To determine the required number of measurements per sample, we study the relationship between the number of measurements per sample and the variation in final scores across independent runs. Specifically, given $K$ measurements for one sample, we compute average scores four times, each averaged from $K$ distinct measurements, denoted as $S_1$, $S_2$, $S_3$, $S_4$. The variation for this sample is defined as $\frac{\max\{S_1,S_2,S_3,S_4\}-\min\{S_1,S_2,S_3,S_4\}}{2}$.

Fig. 5 shows that the variation amplitudes for all three metrics converge when $K = 20$. Therefore, we adopt 20 measurements per sample in all subsequent experiments.

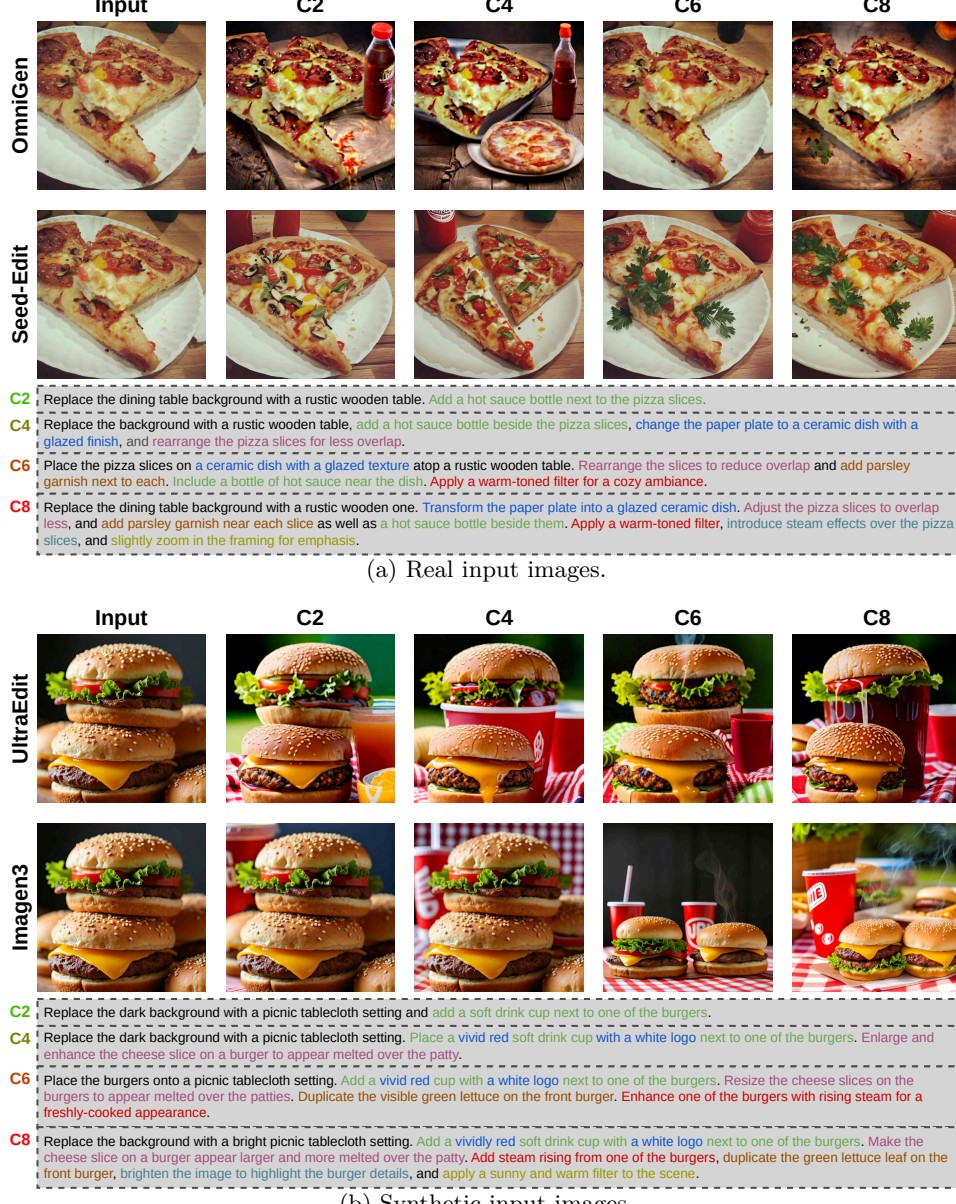

(a) Real input images.

(b) Synthetic input images.

Figure 9: Outputs from open-source and proprietary models via direct editing. See results with more models in the Appendix Figs. 18 and 19.

# 5 Experiments

## 5.1 Experiment Setup

**Dataset.** Our `Complex-Edit` dataset comprises both realistic and synthetic images. Specifically, we select 531 deduplicated images from the EMU-Edit test set (Sheynin et al., 2024) to serve as our realistic image collection. In parallel, we use FLUX.1 (Labs, 2024) to generate an equivalent set of 531 synthetic images based on the captions associated with these real-world images, also sourced from EMU-Edit. Following Sec. 2, we create editing instructions with the complexity ranging from $C_1$ to $C_8$ for these images.

**Models.** Our experiments involve five advanced instruction-based image editing models: three open-source models (UltraEdit (Zhao et al., 2024), OmniGen (Xiao et al., 2024), AnyEdit (Yu et al., 2024)) and three proprietary models (Imagen3 (Baldridge et al., 2024), SeedEdit (Shi et al., 2024), GPT-Image-1 (OpenAI, 2025)). We evaluate each model on both realistic and synthetic images using editing instructions spanning

Table 4: Performance comparison between $C_1$ and $C_8$ on images. All models almost consistently underperform on all metrics when the complexity increases from $C_1$ to $C_8$.

Table 5: Real input images

| Model | $C_1$ | | | | $C_8$ | | | | $C_8 - C_1$ | | | |
|---|---|---|---|---|---|---|---|---|---|---|---|---|
| | $IF$ | $IP$ | $PQ$ | $O$ | $IF$ | $IP$ | $PQ$ | $O$ | $\Delta_{IF}$ | $\Delta_{IP}$ | $\Delta_{PQ}$ | $\Delta_O$ |
| UltraEdit | 7.13 | 7.76 | 7.45 | 7.45 | 6.56 | 5.93 | 7.29 | 6.59 | -0.57 | -1.82 | -0.16 | -0.85 |
| OmniGen | 6.76 | 8.69 | 7.99 | 7.82 | 6.25 | 6.42 | 7.55 | 6.74 | -0.50 | -2.27 | -0.45 | -1.07 |
| AnyEdit | 5.94 | 8.50 | 6.78 | 7.07 | 1.61 | **8.15** | 7.25 | 5.67 | -4.33 | -0.34 | +0.47 | -1.40 |
| Imagen3 | 7.67 | 8.93 | 7.90 | 8.17 | 7.56 | 6.55 | 7.68 | 7.27 | -0.11 | -2.38 | -0.22 | -0.90 |
| SeedEdit | 9.31 | 9.01 | 8.71 | 9.01 | 8.50 | 6.91 | 8.74 | 8.05 | -0.81 | -2.10 | +0.02 | -0.96 |
| GPT-Image-1 | **9.80** | **9.79** | **9.25** | **9.61** | **9.61** | 8.06 | **9.39** | **9.02** | -0.18 | -1.73 | +0.14 | -0.59 |
| Average | 7.77 | 8.78 | 8.02 | 8.19 | 6.68 | 7.00 | 7.98 | 7.22 | -1.08 | -1.77 | -0.03 | -0.96 |

Table 6: Synthetic input image.

| Model | $C_1$ | | | | $C_8$ | | | | $C_8 - C_1$ | | | |
|---|---|---|---|---|---|---|---|---|---|---|---|---|
| | $IF$ | $IP$ | $PQ$ | $O$ | $IF$ | $IP$ | $PQ$ | $O$ | $\Delta_{IF}$ | $\Delta_{IP}$ | $\Delta_{PQ}$ | $\Delta_O$ |
| UltraEdit | 7.82 | 8.61 | 9.02 | 8.48 | 6.51 | 6.83 | 8.72 | 7.35 | -1.30 | -1.78 | -0.30 | -1.13 |
| OmniGen | 7.80 | 9.34 | 9.05 | 8.73 | 6.49 | 7.47 | 8.74 | 7.57 | -1.31 | -1.87 | -0.31 | -1.16 |
| AnyEdit | 5.79 | 9.17 | 8.61 | 7.86 | 1.47 | **8.73** | 8.72 | 6.31 | -4.32 | -0.44 | +0.11 | -1.55 |
| Imagen3 | 8.27 | 9.25 | 8.86 | 8.79 | 7.97 | 7.54 | 8.88 | 8.13 | -0.30 | -1.72 | +0.03 | -0.66 |
| SeedEdit | 9.33 | 9.60 | 9.29 | 9.41 | 8.25 | 7.94 | 9.20 | 8.46 | -1.08 | -1.66 | -0.10 | -0.95 |
| GPT-Image-1 | **9.82** | **9.79** | **9.55** | **9.72** | **9.42** | 8.23 | **9.58** | **9.08** | -0.41 | -1.56 | +0.03 | -0.64 |
| Average | 8.14 | 9.29 | 9.06 | 8.83 | 6.69 | 7.79 | 8.97 | 7.82 | -1.45 | -1.50 | -0.09 | -1.02 |

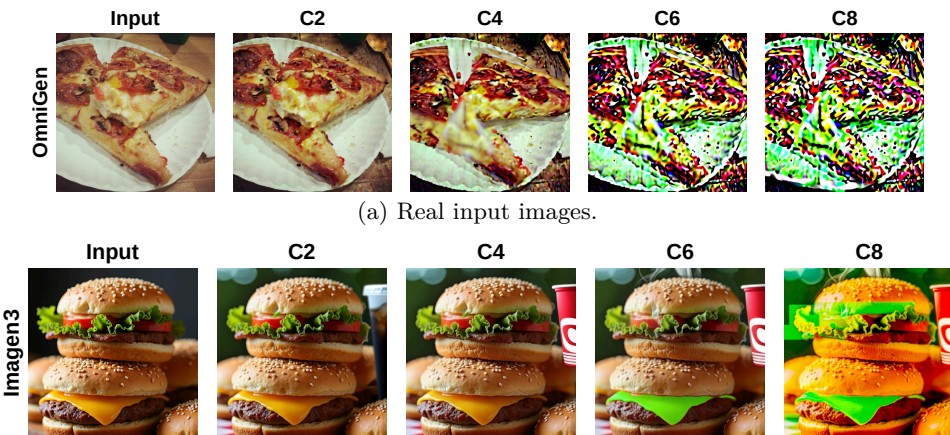

(a) Real input images.

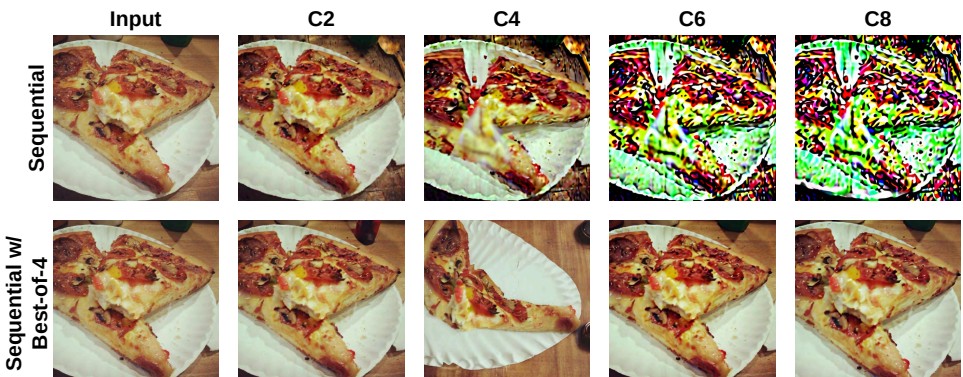

(b) Synthetic input images.

Figure 10: Qualitative results of sequential editing. The complexity level ranges from $C_1$ to $C_8$. It can be seen that $IF$ and $PQ$ are severely hurt by the growing instruction complexity. The instructions are the same ones in Fig. 9. Results with more models are shown in Appendix Figs. 20 and 21.

Figure 11: Qualitative results of sequential editing with and without Best-of-4 with OmniGen on real input images. The complexity level is at $C_8$. This shows that sequential editing can benefit a lot from Best-of-$N$ in terms of $IF$ and $PQ$. The instructions are the same ones in Fig. 9a. Qualitative results with more models are shown in Appendix Fig. 22.

complexity levels from $C_1$ to $C_8$. Note that, owing to usage policies for the proprietary models, results are reported on approximately 60% of the images for Imagen3, 95% for SeedEdit and 90% for GPT-Image-1. All the evaluation are performed by GPT-4o by default. Although Sec. 4 demonstrates that GPT-4o's evaluations are highly aligned with human preferences, it is important to note that a noticeable discrepancy between GPT-4o's judgments and human evaluations still remains. In addition to GPT-4o, we also provide evaluation results by Gemini-2.5-Pro (Comanici et al., 2025) in Appendix Sec. D.

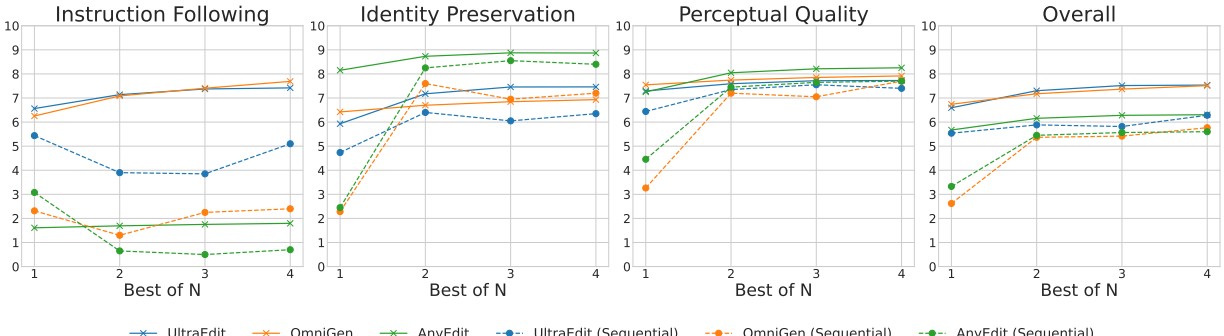

Figure 12: Direct and Sequential Editing results w/ Best-of-$N$ of open-source models on real input images. The complexity level is at $C_8$. This shows that Best-of-$N$ can improve $IF$ and $PQ$, especially for sequential editing. However, sequential editing with Best-of-$N$ can still barely surpass direct editing without Best-of-$N$.

## 5.2 Effect of Increasing Complexity

Fig. 6 illustrates the overall performance trends as instruction complexity increases. In general, higher complexity leads to a consistent decline in Identity Preservation across models, while Instruction Following and Perceptual Quality tend to fluctuate depending on the specific model. For instance, although AnyEdit exhibits a substantial drop in Instruction Following as complexity increases, its Perceptual Quality improves moderately. Additionally, proprietary models outperform open-source alternatives on both realistic and synthetic images, and the performance gap becomes more pronounced at higher complexity levels. Specifically, GPT-Image-1 significantly outperforms all other models at the hardest instruction complexity level $C_8$, particularly in terms of Instruction Following and Perceptual Quality.

Furthermore, both the GPT-4o's and Gemini-2.5-Pro's evaluation in Tab. 4 and Appendix Tabs. 7 and 8 show that the performance drops of open-source models are not notably more severe than those of proprietary models other than GPT-Image-1. For example, Tab. 5 shows that when editing real images, all open-source models have drops around 1.00 in Overall score while the Overall score's drop of Imagen3 and SeedEdit are 0.90 and 0.95 respectively, implying that stronger models may not be more resilient towards the negative impact of increased instruction complexity. Sample qualitative results are shown in Fig. 9.

## 5.3 Curse of Synthetic data

Our qualitative analysis reveals an intriguing phenomenon: when applying extremely complex editing instructions ($C_8$) to real input images, the resulting outputs frequently lose their photorealism and adopt a distinctly synthetic aesthetic. Notably, UltraEdit is particularly susceptible to this effect compared to OmniGen and AnyEdit, as illustrated in Fig. 7. We attribute this trend to the composition of UltraEdit's training data, which contains a significantly higher proportion of synthetic images than the datasets used by OmniGen and AnyEdit (see Tab. 3). This observation aligns with the "curse of synthetic data" phenomenon discussed in (Gerstgrasser et al., 2024).

This phenomenon also extends to highly capable proprietary models, including SeedEdit, Imagen3, and even GPT-Image-1. Specifically, as illustrated in Fig. 8, the edited images produced by GPT-Image-1 completely lose their photorealism, even though its overall performance surpasses that of all other models evaluated. This observation may indicate that, although the training sources for Imagen3 and GPT-Image-1 are undisclosed, these models likely incorporate synthetic data into their training sets, which in turn contributes to their impressive performance as well as the emergence of a distinctly synthetic aesthetic. Moreover, the identified "curse of synthetic data" underscores the risk that repeated reliance on synthetic images in training and evaluation may gradually erode photorealism and authenticity. This stresses the necessity of incorporating safeguards to mitigate possible self-reinforcing cycles, such as balancing synthetic datasets with high-quality real-world examples or implementing rigorous human-in-the-loop verification steps. We detail the evaluation protocol for this phenomenon in Appendix Sec. E.

### 5.4 Test-Time Scaling Approaches

### 5.4.1 CoT-like Sequential Editing

Our data pipeline composes complex instructions by combining sequences of atomic editing operations, suggesting that sequential application of these atomic steps should be equivalent to executing the complex instruction directly. Moreover, this way of decomposing a complex task and then sequentially executing it has been proven effective in both language generation (Wei et al., 2022) and T2I generation (Guo et al., 2025).

Formally, given an input image $x$, a complex instruction $T$ at complexity level $C_i$, and a corresponding sequence of atomic instructions $t_1, \ldots, t_i$, we define the outputs for direct and sequential editing as follows: $y_{\text{direct}} = f(x, T)$, and $y_{\text{sequence}} = f^i(x, \{t_1, \ldots, t_i\}) = f(f(f(\ldots f(x, t_1) \ldots, t_{i-1}), t_i))$.

We evaluate sequential editing on both real and synthetic images. Our results (summarized in Fig. 6 and illustrated qualitatively in Fig. 10) reveal that sequential editing yields a steady decline in performance across all three metrics, with accumulating visual artifacts and distortions—even for strong proprietary models such as Imagen3, SeedEdit and GPT-Image-1. Notably, AnyEdit demonstrates improved Instruction Following with sequential editing compared to direct editing, although this improvement is offset by a reduction in Identity Preservation.

### 5.4.2 Best-of-$N$

We also experiment with Best-of-N, which is another simple test-time-scaling method commonly used for text-to-image generation (Guo et al., 2025; Ma et al., 2025). For direct editing, we generate $N$ candidate outputs for each input image and select the one with the highest overall score $O$. For sequential editing, we generate $N$ candidates at each step and choose the candidate with the highest overall score $O$ to proceed to the next editing operation. When evaluating an intermediate candidate at the $i$-th step during sequential editing, instead of considering it as the output of a single-step edit produced with the predecessor output and an atomic instruction, $e.g.$ $f(y_{i-1}, t_i)$ with $y_{i-1} = f^{i-1}(x, \{t_1, \ldots, t_{i-1}\})$, we evaluate it as the output from the original input image with a compound instruction $T_i$. This compound instruction $T_i$ has $C_i$ complexity and encompasses all the atomic instructions up to the $i$-th step: $\{t_1, \ldots, t_i\}$. The rationale here is that the intermediate result should best represent the cumulative effects of all preceding atomic steps on the original image, ensuring the final output best reflects all atomic steps and consequently the final compound instruction.

Our evaluations with open-source models on real-life images (see Fig. 12) indicate that increasing $N$ gradually improves direct editing performance across all metrics. For sequential editing, even a Best-of-$N$ strategy with $N = 2$ produces significant gains in Identity Preservation and Perceptual Quality; however, the improvement in Instruction Following is less consistent. Qualitative comparisons are provided in Fig. 11.

## 6 Related Works

**Instruction-Based Image Editing** InstructPix2Pix (Brooks et al., 2023) introduced instruction-based image editing, where images are modified through text instructions, improving user interaction. This required a large dataset created using fine-tuned GPT-3 (Achiam et al., 2023) and the Prompt-to-Prompt (Hertz et al., 2022) Diffusion Model, but editing was limited by these models' quality. Research explored new methods: MagicBrush (Kawar et al., 2023) used curated annotations, HQ-Edit (Hui et al., 2025) utilized advanced language models with text-to-image systems for better datasets, MGIE (Fu et al., 2023) and SmartEdit (Huang et al., 2024) applied multilevel language learning for more precise edits, and UltraEdit (Zhao et al., 2024) used advanced masking for detailed manipulation. OmniGen (Xiao et al., 2024) expanded model architecture and data with a Transformer framework for processing text and visuals and inferring edits. UIP2P (Simsar et al., 2024) introduced cyclic editing for unsupervised image editing without paired training images. These methods have significantly advanced instruction-based image editing.

# 7 Conclusion

In this work, we introduced `Complex-Edit`, a comprehensive benchmark designed to systematically evaluate instruction-based image editing models across varying levels of instruction complexity. Through extensive experiments, `Complex-Edit` enable us to uncover multiple key insights regarding the limitations and capabilities of current models. We hope this benchmark will drive future advancements in developing more powerful instruction-based image editing models, especially those with test-time scaling ability.

## Acknowledgement

We would like to thank Google Cloud Research Credits Program, and the Microsoft Accelerating Foundation Models Research Program for supporting our computing needs.

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

# A  The Hierarchy of Editing Complexity Types

To better contextualize the contribution of `Complex-Edit` and clarify our definition of instruction complexity, we propose a hierarchy of editing complexity types for instruction-based image editing.

**Tier 1 (T1): Simple Edit.**  This tier primarily operates on a *perceptual* basis, comprising single-step operations that correspond to the manipulation of a specific visual attribute or object (*e.g.*, "add a cat," "dim the background"). The principal technical challenge is **visual grounding** and **regeneration**—that is, accurately locating the target pixels and generating consistent textures. Most existing datasets and benchmarks focus on assessing this foundational capability.

**Tier 2 (T2): Semantic-Complex Edit.**  This tier introduces *structural and linguistic* challenges. The difficulty stems not just from generating pixels, but from processing dense instructions and resolving the inter-relationships between multiple visual elements. This is the primary focus of `Complex-Edit`.

- **Compositional Complexity.** The instruction contains multiple atomic operations, creating a high *compositional load.* The core challenge is satisfying multiple constraints simultaneously (*e.g.*, "Add a red car on the left and a blue bicycle on the right") while maintaining **attribute binding** and **disentanglement**. Successful editing requires the model to prevent semantic leakage (*e.g.*, ensuring the car does not absorb the color blue) among the dense set of constraints.

- **Contextual Complexity.** The instruction relies on *linguistic context*, such as pronouns, ellipsis, or multi-turn interaction history. The challenge lies in **coreference resolution** (*e.g.*, "Add a dog. Make *it* look angry") and **pragmatic interpretation** (*e.g.*, "Give the man on the left a pair of glasses. Same for the one on the right"), where the requested edit is implied by context rather than an self-contained description.

**Tier 3 (T3): Inferential-Complex Edit.**  Unlike Tier 2, where instructions are concrete and descriptive, this tier involves implicit or abstract directives that typically require *cognitive-level* processing. The model maybe presented with a high-level goal rather than a checklist of changes. The core challenge shifts from execution to **latent planning**, as the instruction does not directly describe the final visual state.

- **Reasoning Complexity.** The instruction implies a *causal relationship*. Determining *what* to edit requires **logical derivation**. For instance, "Add a support beam where the bridge is structurally weak" implies a multi-step process: analyzing the force distribution to locate potential weakness → placing the beam. The edit is contingent on understanding the scene's physical logic rather than text matching.

- **Knowledge Complexity.** The instruction relies on specific *world knowledge* or factual relations that have no direct visual correspondence in the text. The challenge involves translating a non-visual entity into specific visual attributes via **knowledge retrieval**. For example, in the instruction "Change the animal to the natural predator of the seal in the Arctic," the model cannot simply map the keyword "predator" to a specific texture; it necessitates establishing the factual link: arctic predator of seal → polar bear.

**Tier 4 or Higher (≥ T4): Future Frontiers.**  Defining the upper tiers of this hierarchy remains an open area of exploration. We anticipate that these tiers will necessitate capabilities that transcend the current definition of image editing, such as **long-term planning** (e.g., in multi-step agentic workflows), **iterative self-reflection**.

**Complexity Tier vs. Execution Difficulty.**  It is essential to distinguish between the *cognitive tier* of a task and its *execution difficulty.* While **T3 (Inferential Complexity)** conceptually represents a higher level of intelligence requiring latent planning, **T2 (Semantic Complexity)** rigorously tests the model's capacity to process dense, structured information involving intricate compositional and contextual

dependencies. Intuitively, executing a high-density T2 instruction (e.g., $C_8$)—which requires simultaneously satisfying numerous specific constraints—can impose a heavier processing load than a T3 task involving simple logical deduction. In this high-load regime, models are prone to "losing focus" or omitting details due to the overwhelming semantic density, a vulnerability that `Complex-Edit` specifically targets.

## B  Atomic Editing Operations

Here we present the descriptions for all the 24 atomic operation types. These operation types are also fed to GPT-4o during Stage #1 (Sequence Generation) of our data pipeline, as discussed in Sec. 2.1.

- **Object Manipulation and Transformation**
  - *Add an Object*: Insert a new element into the image.
  - *Remove an Object*: Eliminate an existing element from the image.
  - *Replace an Object*: Swap one element with another.
  - *Move an Object*: Change the position of an existing element within the image.
  - *Resize an Object*: Adjust the size of an existing element.
  - *Rotate an Object*: Rotate an element to a specified angle.
  - *Duplicate an Object*: Create a copy of an existing element.

- **Color and Tone Adjustments**
  - *Change Color*: Replace the color of an element with a specified color.
  - *Apply Filter/Weather*: Add a color filter or weather effect to the entire image or specific parts.

- **Texture and Material Adjustments**
  - *Change Texture*: Apply a texture to an element (*e.g.*, change from metal to wood).

- **Background and Environment**
  - *Change Background*: Replace the background with a different scene or color.

- **Lighting and Shadows**
  - *Adjust Lighting*: Change the overall lighting or lighting of specific elements.

- **Text and Symbols**
  - *Add Text*: Insert text into the image.
  - *Remove Text*: Eliminate existing text from the image.
  - *Change Text Properties*: Modify font, color, size, or position of existing text.

- **Pose and Expression**
  - *Change Pose*: Modify the stance or posture of a person or object.
  - *Change Facial Expression*: Alter the facial expression of a character.

- **Composition and Cropping**
  - *Crop Image*: Adjust the framing of the image by removing outer areas.
  - *Reframe Composition*: Change the focus or arrangement of elements within the image.
  - *Zoom In/Out*: Adjust the zoom level to focus on specific elements or show a broader view.

- **Special Effects**
  - *Add Special Effects*: Introduce effects like glow, motion blur, or lens flare.
  - *Remove Special Effects*: Eliminate existing special effects from the image.
  - *Add Particles*: Insert particles like dust.
  - *Remove Particles*: Remove existing particles from the image.

# C   Prompts for Evaluation

## C.1   Rubrics

In the rubric for each metric we propose, we specify the detailed textual description for every score from 0 to 10, and most scores have a textual example to reduce ambiguity. All the rubrics for evaluation in are listed in Fig. 13.

---

**Rubric for Instruction Following.**

* 10 (Perfect Instruction Following): All the required changes occur in the output image.
* 9 (Near Perfect Instruction Following with negligible deviations): Almost all instructed changes are present but negligible deviations exist (e.g., a tiny color variation such as the cat in the image is now black but the ears are grey).
* 7-8 (Strong Instruction Following with minor deviations): Most required changes are applied accurately. Minor deviations exist but do not substantially alter the intended modification (e.g., a car is changed to blue as instructed, but the reflection on its surface still contains a red tint).
* 5-6 (Moderate Instruction Following with noticeable deviations): The output reflects an attempt to follow instructions but with moderate errors (e.g., adding a required element but with incorrect attributes like color or shape).
* 3-4 (Weak Instruction Following with major deviations): Most required modifications are missing, incorrect, or only vaguely implemented. Significant elements from the instruction are misrepresented (e.g., when instructed to add a hat, a small, barely visible accessory is added to the head, but it does not resemble a proper hat).
* 1-2 (Minimal Instruction Following with severe deviations): A vague attempt is made, but the required modifications are either incorrect or so minimal that they do not fulfill the instruction (e.g., the instruction asks to remove a person from the image, but they are still visible, just slightly blurred or faded instead of being properly erased.).
* 0 (Complete failed Instruction Following): The output image is entirely unrelated to the instruction.

(a) Instruction Following.

---

**Rubric for Identity Preservation.**

* 10 (Perfect Identity Preservation): All key elements that should remain unchanged are completely preserved and indistinguishable from the input (e.g., a person's face, expression, and proportions remain completely unchanged except for the required edits).
* 9 (Near Perfect Identity Preservation with negligible distortion): Key elements that should remain unchanged are preserved with negligible distortion (e.g., A person's face is identical except for a tiny, imperceptible variation in hair texture).
* 7-8 (Strong Identity Preservation with minor distortion): Small details of the key elements may have changed, but they do not significantly disrupt the overall identity (e.g., a pet's fur pattern remains mostly accurate, but a minor detail like a stripe or spot is different).
* 5-6 (Moderate Identity Preservation with noticeable distortion): Most of the key elements remain recognizable but with noticeable distortions (e.g., the instruction asks to change a car's color, but the car's shape or size is modified along with the color).
* 3-4 (Weak Identity Preservation with major distortion): Key elements maintain a general resemblance but noticeable changes are present (e.g., the instruction asks to brighten the sky, but additional buildings in the background appear or disappear).
* 1-2 (Minimal Identity Preservation with severe distortion): Most key elements are significantly altered or replaced. The key elements in the output retain only minor aspects of the original, but major features are incorrect (e.g., a person's face is still a human face, but it no longer resembles the original person at all).
* 0 (Complete failed Identity Preservation): All key elements that should remain unchanged are altered, distorted, or missing.

(b) Identity Preservation.

---

**Rubric for Perceptual Quality.**

* 10 (Perfect Perceptual Quality): The image appears flawlessly natural, and all objects are seamlessly integrated into the environment with consistent lighting and shadows. There is no visual artifact at all.
* 9 (Near Perfect Perceptual Quality with negligible incoherence): The image is very close to perfect, but a tiny, almost imperceptible inconsistency exists. Seamless integration, but one might notice an extremely subtle flaw. (e.g., a person added to a group photo blends in perfectly, but upon close examination, their shadow is slightly softer than others.)
* 7-8 (Strong Perceptual Quality with minor incoherence): Minor incoherence and artifacts are present but they do not significantly detract from the overall harmony. (e.g., a sunset scene where the added reflections on water are slightly off in intensity, but the image still looks highly realistic.)
* 5-6 (Moderate Perceptual Quality with noticeable incoherence): There is noticeable visual artifacts affecting the image's harmony. Lighting and shadows may be misaligned or inconsistent. (e.g., an animal is distorted in size or shape, making it appear out of place in the scene.)
* 3-4 (Weak Perceptual Quality with major incoherence): Disharmonious elements are prominent, greatly disturbing the visual harmony. (e.g., an animal's shape or a person's face is greatly distorted, only showing some resemblance of the animal species or a human face.)
* 1-2 (Minimal Perceptual Quality with severe incoherence): The whole scene is distorted, making it difficult to recognize the objects or subjects in the image.
* 0 (Complete failed Perceptual Quality): The image is completely random and makes no sense at all.

(c) Perceptual Quality.

Figure 13: Rubrics for evaluating Instruction Following, Identity Preservation and Perceptual Quality.

### C.2 System Prompts

The system prompts that we use for evaluation are provided in Fig. 14.

---

**System Prompt for Instruction Following and Identity Preservation.**

You are required to evaluate the result of an instruction-based image editing model.
Given an input image, an output image and a text instruction, you are required to access the output image based on whether the changes made to the input image align with the text instruction.

You are required to give two integer scores in [0, 10] based on the following criteria:
1. Instruction Following: whether the required changes occur in the output image, regardless of whether unnecessary changes are also made. 10 means that all the changes required by the instruction occur in the output image, 0 means that no changes required by the instruction occur in the output image.
2. Identity Preservation: whether elements that should not be changed stay the same in the output image, regardless of whether required changes occur. 10 means that no unnecessary changes occur in the output image, 0 means that all elements in the input image that should be kept the same are changed in the output image.

Here is the detailed rubric for Instruction Following:
`<INSTRUCTION_FOLLOWING_RUBRIC>`

Here is the detailed rubric for Identity Preservation:
`<IDENTITY_PRESERVATION_RUBRIC>`

Note that these two scores should be graded independently, and a low score for one criterion should not affect the score for the other criterion. For example, an output image that is identical to the input image should have an Instruction Following score of 0, but an Identity Preservation score of 10. Also, an output image that has no relevance with the input image should have an Identity Preservation score of 0 unless the instruction specifically orders the model to create a whole different image, but it should not affect the Instruction Following score as long as changes required by the instruction occur in the output.

If the instruction contains several atomic operations, evaluate the Instruction Following for each atomic operation separately and then average the scores as the assessment for Instruction Following.

Explain your reasoning before answering the question.

---

(a) Instruction Following and Identity Preservation.

---

**System Prompt for Perceptual Quality.**

You are required to evaluate a model-generated image.
Given an output, you are required to access the output image's "Perceptual Quality".

You are required to give one integer score in [0, 10] with 0 indicating extreme disharmony characterized by numerous conflicting or clashing elements, and 10 indicating perfect harmony with all components blending effortlessly.

These are the criteria:
1. Consistency in lighting and shadows: The light source and corresponding shadows are consistent across various elements, with no discrepancies in direction or intensity.
2. Element cohesion: Every item in the image should logically fit within the scene's context, without appearing misplaced or extraneous.
3. Integration and edge smoothness: Objects should blend seamlessly into their surroundings, with edges that do not appear artificially inserted or poorly integrated.
4. Aesthetic uniformity and visual flow: The image should not only be aesthetically pleasing but also facilitate a natural visual journey, without abrupt interruptions caused by disharmonious elements.

Here is the detailed rubric for scoring:
`<PERCEPTUAL_QUALITY_RUBRIC>`

---

(b) Perceptual Quality.

Figure 14: System prompt for evaluating Instruction Following, Identity Preservation and Perceptual Quality.

# D    Evaluation Bias due to VLM Evaluators

To avoid evaluation completely relying on GPT-4o, we opt for another VLM evaluator, Gemini-2.5-Pro (Comanici et al., 2025), which is from a distinct family.

Here we present the comparison between evaluation results of direct editing with $C_1$ and $C_8$ instructions, in supplement to the results from GPT-4o evaluation shown in Tab. 4. We only measured once for each sample with Gemini-2.5-Pro due to cost limitation. As we report average scores across the test set, the variance of each score is expected to be $< 0.01$, sufficient for reliability.

The observations based on Tabs. 7 and 8 remain consistent as ones discussed in Sec. 5.2, that at the highest instruction complexity level $C_8$, GPT-Image-1 excels over all other models in Instruction Following and Perceptual Quality; and performance drops in open-source models are comparable to proprietary ones, except for GPT-Image-1.

Table 7: Performance comparison between $C_1$ and $C_8$ on real images. All models almost consistently underperform on all metrics when the complexity increases from $C_1$ to $C_8$. Evaluated by Gemini-2.5-Pro.

| Model | $C_1$ | | | | $C_8$ | | | | $C_8 - C_1$ | | | |
|---|---|---|---|---|---|---|---|---|---|---|---|---|
| | $IF$ | $IP$ | $PQ$ | $O$ | $IF$ | $IP$ | $PQ$ | $O$ | $\Delta_{IF}$ | $\Delta_{IP}$ | $\Delta_{PQ}$ | $\Delta_O$ |
| UltraEdit | 6.63 | 5.26 | 5.94 | 5.94 | 5.25 | 3.37 | 4.98 | 4.53 | -1.38 | -1.89 | -0.96 | -1.41 |
| OmniGen | 6.43 | 7.44 | 6.82 | 6.90 | 5.17 | 4.35 | 5.47 | 5.00 | -1.26 | -3.09 | -1.35 | -1.90 |
| AnyEdit | 5.26 | 7.31 | 5.50 | 6.02 | 1.01 | 7.81 | 6.51 | 5.11 | -4.25 | +0.51 | +1.01 | -0.91 |
| Imagen3 | 7.25 | **8.32** | 7.71 | 7.76 | 6.70 | 4.65 | 6.53 | 5.96 | -0.55 | -3.67 | -1.18 | -1.80 |
| SeedEdit | 9.17 | 6.29 | 7.96 | 7.81 | 7.67 | 3.83 | 7.69 | 6.40 | -1.50 | -2.46 | -0.27 | -1.41 |
| GPT-Image-1 | **9.78** | 8.06 | **9.10** | **8.98** | **9.45** | **5.04** | **9.25** | **7.92** | -0.33 | -3.01 | +0.15 | -1.06 |
| Average | 7.42 | 7.11 | 7.17 | 7.24 | 5.88 | 4.84 | 6.74 | 5.82 | -1.54 | -2.27 | -0.43 | -1.42 |

Table 8: Performance comparison between $C_1$ and $C_8$ on synthetic images. All models almost consistently underperform on all metrics when the complexity increases from $C_1$ to $C_8$. Evaluated by Gemini-2.5-Pro.

| Model | $C_1$ | | | | $C_8$ | | | | $C_8 - C_1$ | | | |
|---|---|---|---|---|---|---|---|---|---|---|---|---|
| | $IF$ | $IP$ | $PQ$ | $O$ | $IF$ | $IP$ | $PQ$ | $O$ | $\Delta_{IF}$ | $\Delta_{IP}$ | $\Delta_{PQ}$ | $\Delta_O$ |
| UltraEdit | 7.31 | 6.82 | 8.50 | 7.55 | 4.97 | 4.70 | 7.69 | 5.79 | -2.34 | -2.12 | -0.81 | -1.76 |
| OmniGen | 7.46 | 8.68 | 8.52 | 8.22 | 5.15 | 6.19 | 7.68 | 6.34 | -2.31 | -2.48 | -0.84 | -1.88 |
| AnyEdit | 5.31 | 8.68 | 7.98 | 7.32 | 0.78 | **8.83** | 8.33 | 5.98 | -4.53 | +0.15 | +0.35 | -1.34 |
| Imagen3 | 7.64 | 8.43 | 8.49 | 8.19 | 7.11 | 6.54 | 8.37 | 7.34 | -0.53 | -1.90 | -0.12 | -0.85 |
| SeedEdit | 9.05 | 8.36 | 8.96 | 8.79 | 6.97 | 6.86 | 8.63 | 7.49 | -2.08 | -1.50 | -0.33 | -1.30 |
| GPT-Image-1 | **9.69** | **8.68** | **9.40** | **9.26** | **9.09** | 6.17 | **9.54** | **8.27** | -0.60 | -2.52 | +0.15 | -0.99 |
| Average | 7.74 | 8.27 | 8.64 | 8.22 | 5.68 | 6.55 | 8.37 | 6.87 | -2.07 | -1.73 | -0.27 | -1.35 |

# E   Human Evaluation of Curse of Synthetic Data

To isolate the influence of Perceptual Quality on the assessment of an image's photorealistic style, we randomly selected 50 input samples at $C_8$ complexity level for which all evaluated models attained a $PQ \geq 8$. For each selected input, the corresponding output images produced by each model – 300 images in total – were subsequently evaluated by a human rater. The rater was instructed to assign a photorealism score to each image on a discrete ordinal scale ranging from 1 (totally non-photorealistic) to 5 (perfectly photorealistic), as shown in Fig. 15.

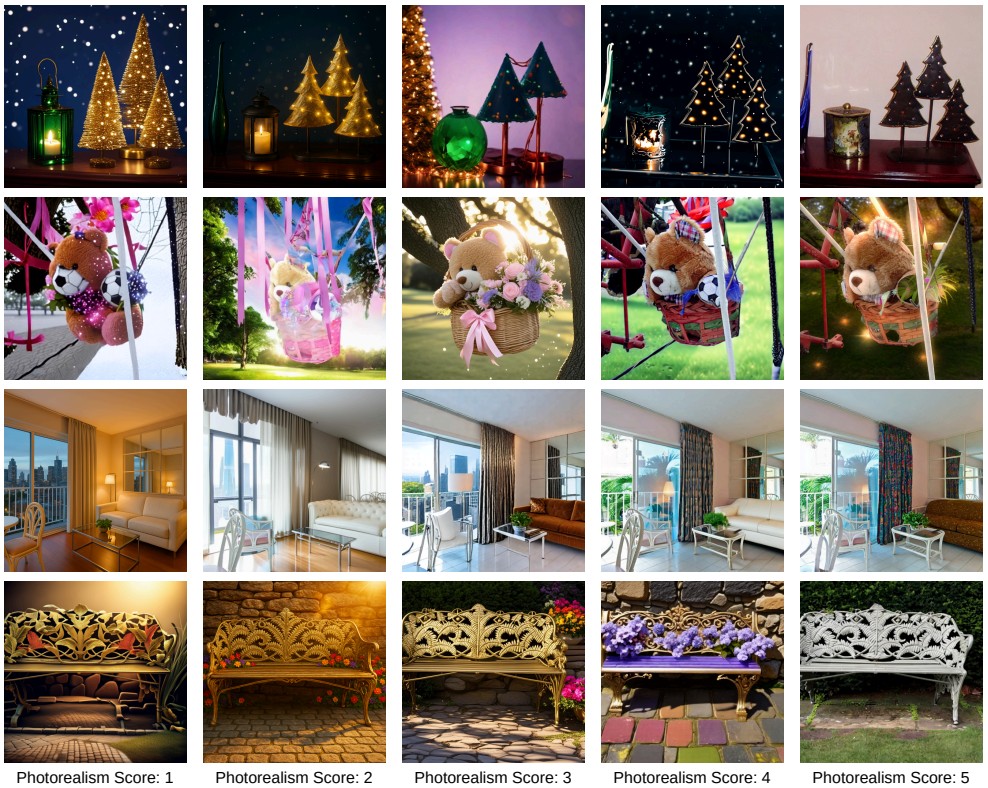

Photorealism Score: 1    Photorealism Score: 2    Photorealism Score: 3    Photorealism Score: 4    Photorealism Score: 5

Figure 15: Examples of human-evaluated photorealism score.

Table 9: Evaluation of realistic style with direct editing on real images with the instruction complexity at $C_8$.

| Model | Prop. of Real Images | Realisim Score |
|---|:---:|:---:|
| UltraEdit | ✓ | 2.84 |
| OmniGen | ✓✓✓ | 3.72 |
| AnyEdit | ✓✓✓ | 4.74 |
| SeedEdit | ✓ | 2.84 |
| Imagen3 | ? | 2.76 |
| GPT-Image-1 | ? | 1.30 |

Given that this evaluation protocol is relatively rudimentary, we report the exact photorealism scores only in Appendix Tab. 9, and omit them from Tab. 3. We rely exclusively on human evaluation rather than VLM-based automatic assessment because, in the context of image editing, the photorealism of an output image must often be judged under prompts that intentionally include fantastical or physically implausible elements (e.g., science-fiction or magical content). Assessing such "photorealism" requires the evaluator to possess not only high-fidelity visual perception, but also counterfactual reasoning capabilities—for instance, the ability to infer "how a unicorn should plausibly appear if unicorns were to exist".

# F  Additional Qualitative Results

Here we present more comprehensive qualitative results.

## F.1  Examples of Evaluation Metrics

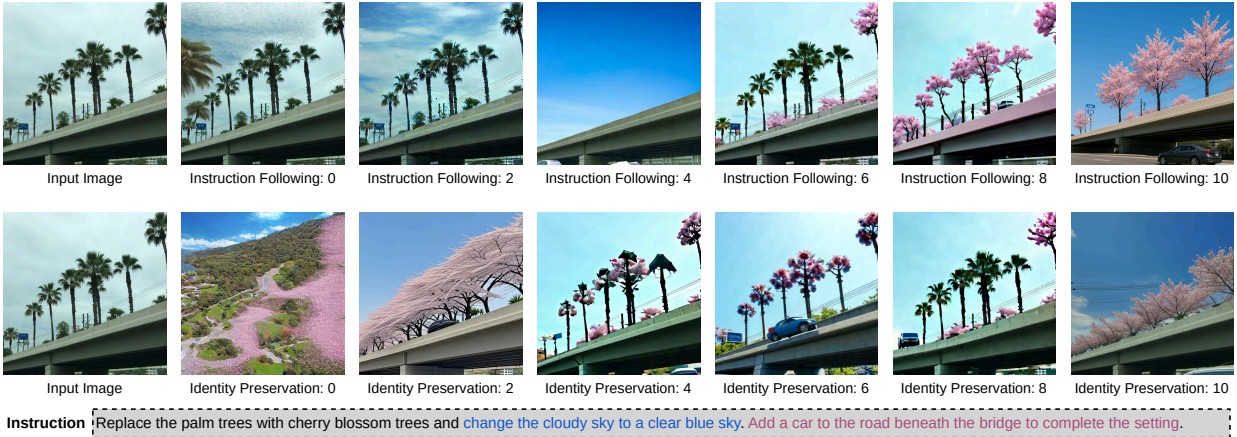

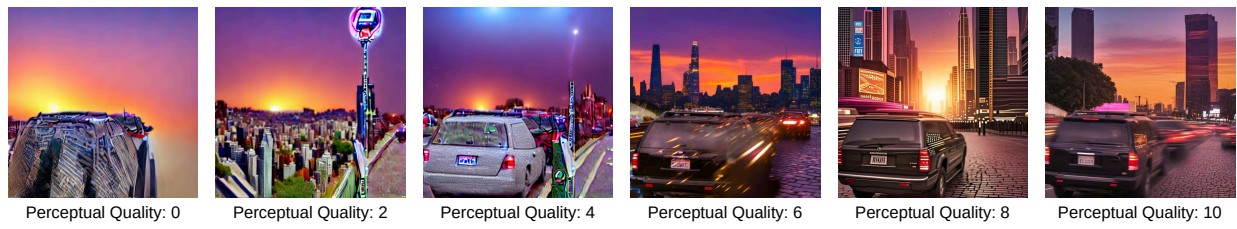

Figure 16: Examples of evaluation results for Instruction Following and Identity Preservation.

Figure 17: Examples of evaluation results for Perceptual Quality.

## F.2  Direct Editing

Additional qualitative results of direct editing with more models are shown in Figs. 18 and 19, indicating that the gap between open-source and proprietary models expands with increasing complexity of the instruction. Also, Identity Preservation and Perceptual Quality continue to drop as instruction complexity increases, while the effect on Instruction Following varies from model to model. One can refer to the discussion in Sec. 5.2 for more detailed insights.

## F.3  Sequential Editing

We illustrate additional qualitative results of sequential editing with more models in Figs. 20 and 21, showing that all three aspects, particularly Identity Preservation and Perceptual Quality, consistently degrade, as visual artifacts and distortions accumulate with an increase in the number of intermediate steps. Even advanced proprietary models, *i.e.* Imagen3 and SeedEdit, demonstrate limitations in maintaining the identity of key elements and the overall quality of output images. More in-depth discussions are given in Sec. 5.4.1.

## F.4  Best-of-N

Additional qualitative results of sequential editing combined with Best-of-4 with more open-source models are shown in Fig. 22. It can been noticed that the Identity Preservation and Perceptual Quality of sequential

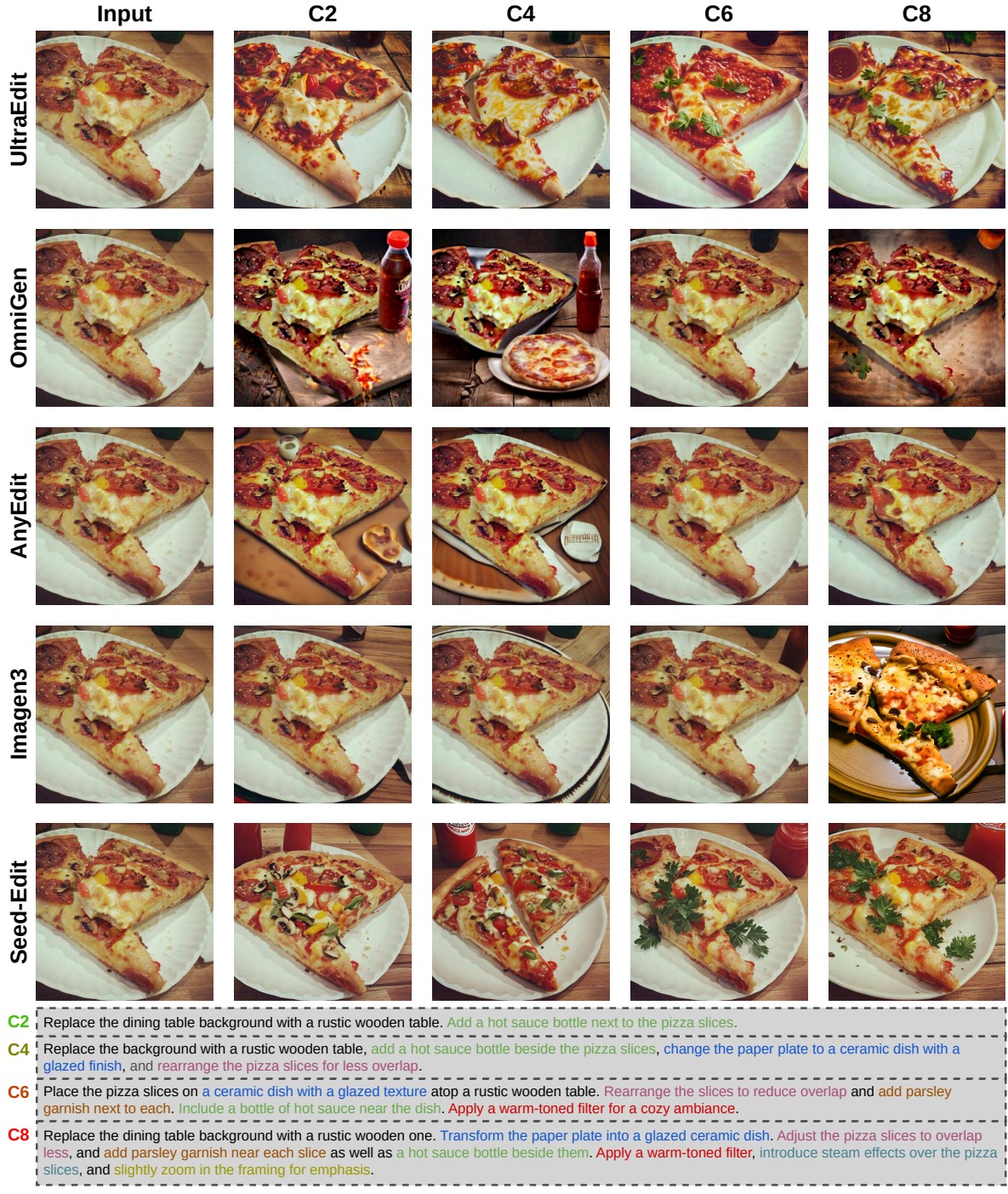

Figure 18: Additional qualitative results with direct editing on real-life images. As instruction complexity increases, the gap between open-source and proprietary models widens, and both Identity Preservation and Perceptual Quality decline, with variable effects on Instruction Following across models. See Sec. 5.2 for more details.

editing's results can be substantially improved with Best-of-4. A more comprehensive analysis of this topic is provided in Sec. 5.4.2.

The Identity Preservation and Perceptual Quality of sequential editing improve significantly with Best-of-4, as analyzed in detail in Sec. 5.4.2.

### F.5 Direct Editing with GPT-Image-1

We provide more qualitative results with GPT-Image-1 via direct editing real-life images in Fig. 23. These results imply that while GPT-Image-1 is capable of generating edited images of exceptional quality, output images with very complex instructions prune to lose the photorealism in the original images. This may imply a lack of real-life images in GPT-Image-1's training data. See Sec. 5.3 for a more in-depth analysis.

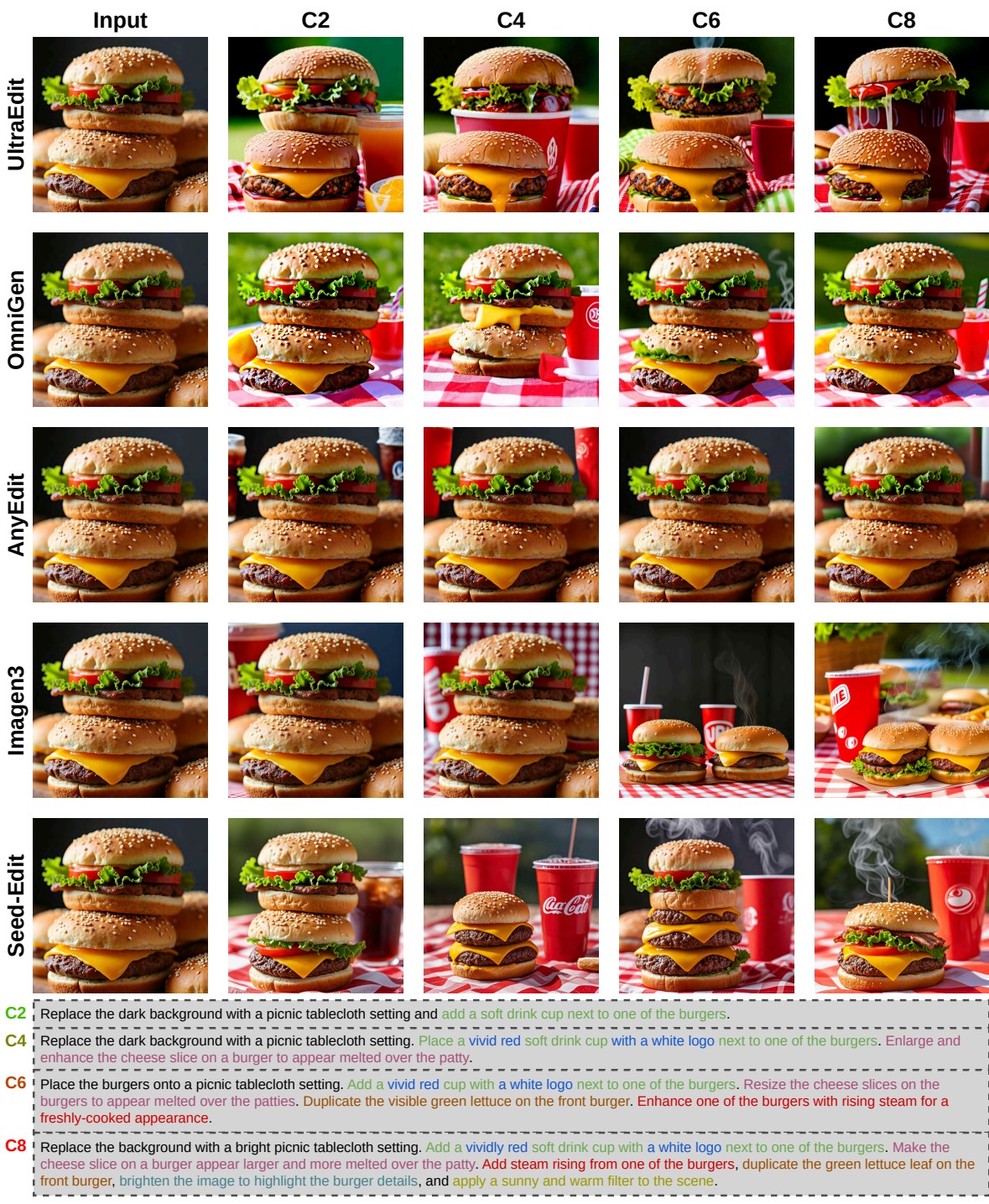

Figure 19: Additional qualitative results with direct editing on synthetic images. As instruction complexity increases, the gap between open-source and proprietary models widens, and both Identity Preservation and Perceptual Quality decline, with variable effects on Instruction Following across models. See Sec. 5.2 for more details.

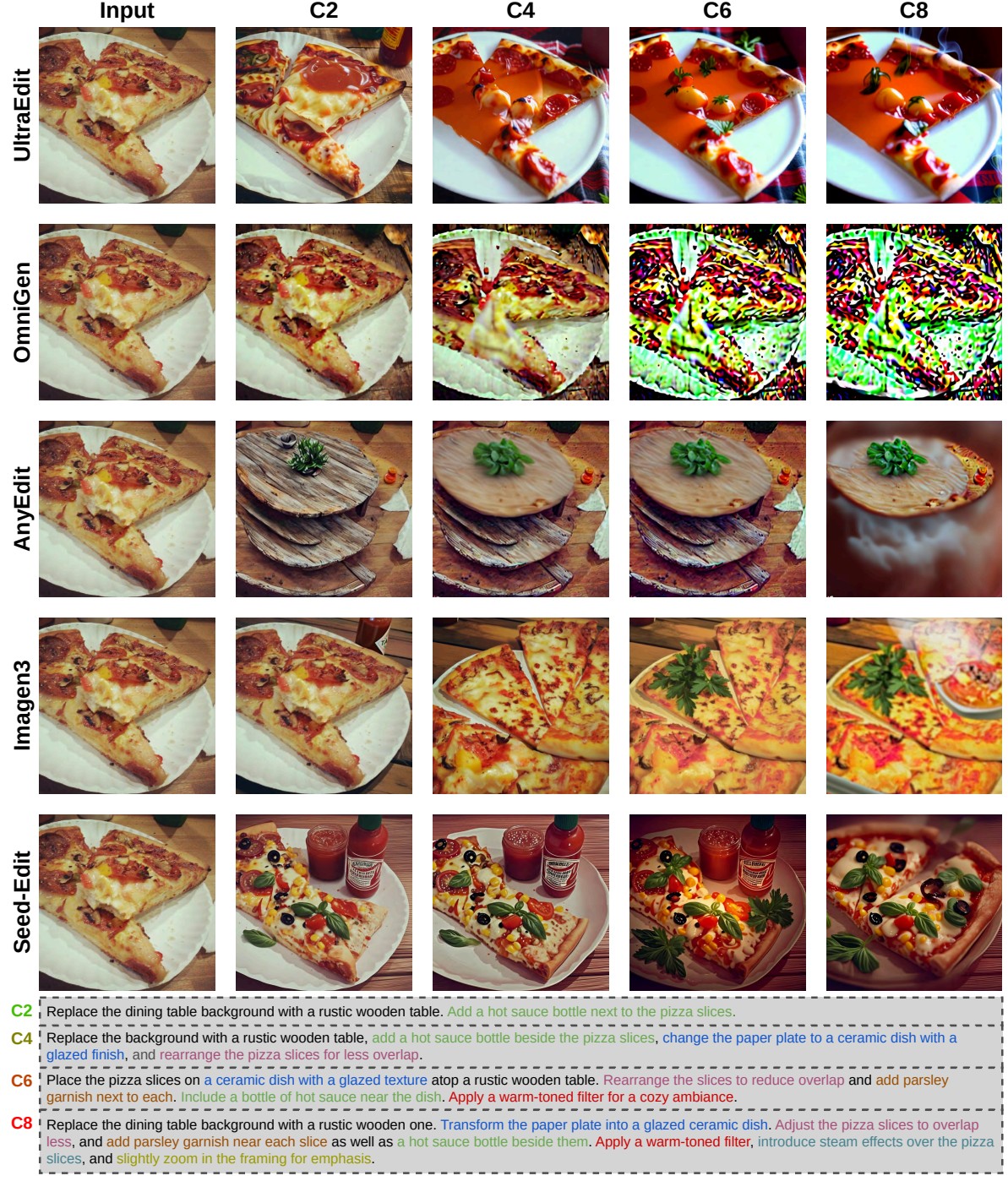

Figure 20: Additional qualitative results with sequential editing on real-life images. Identity Preservation and Perceptual Quality, degrade as visual artifacts and distortions increase with more intermediate steps. Even advanced proprietary models, *i.e.* Imagen3 and SeedEdit, struggle to maintain element identity and quality in output images. More in-depth analysis is in Sec. 5.4.1.

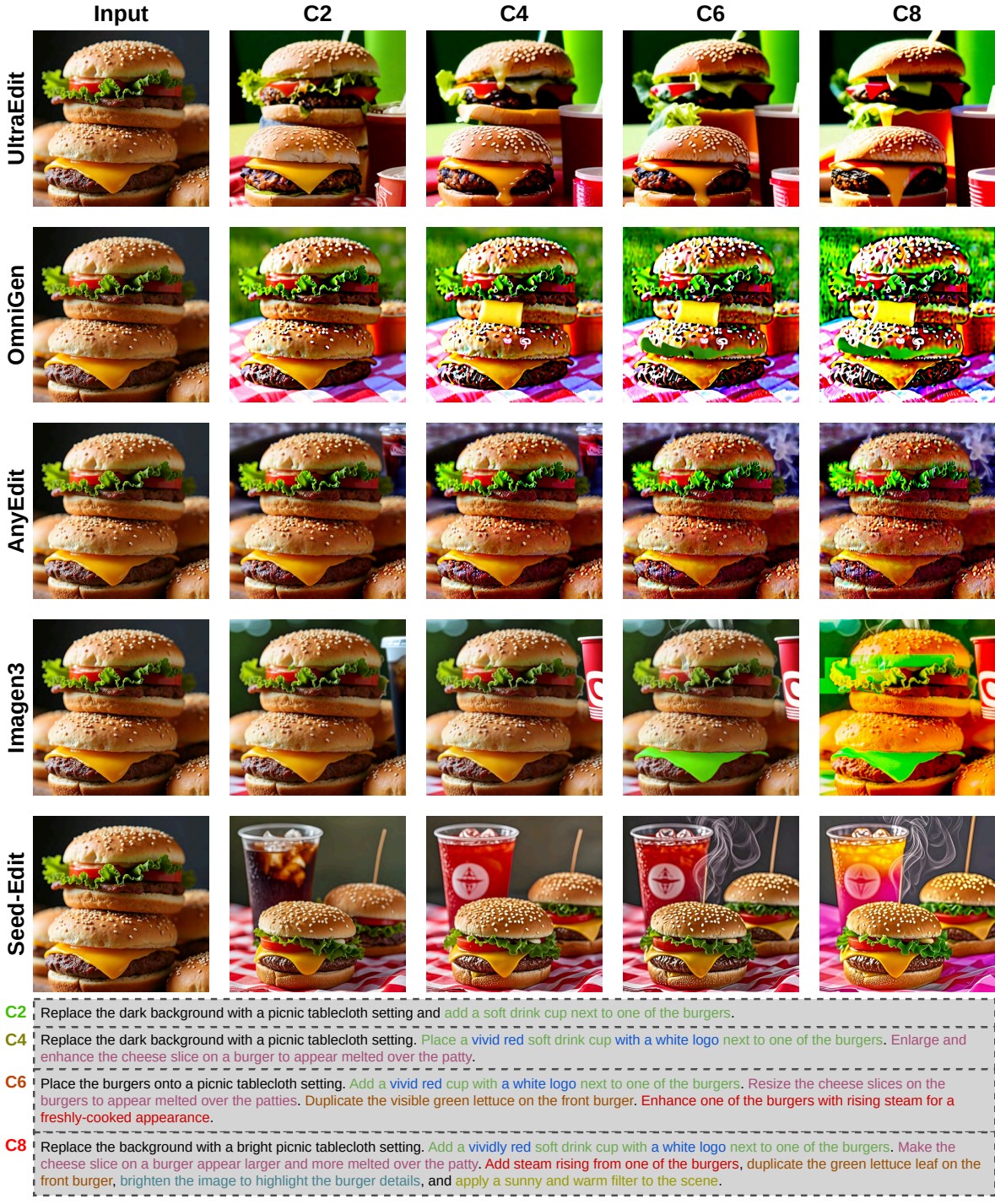

Figure 21: Additional qualitative results with sequential editing on synthetic images. Identity Preservation and Perceptual Quality, degrade as visual artifacts and distortions increase with more intermediate steps. Even advanced proprietary models, *i.e.* Imagen3 and SeedEdit, struggle to maintain element identity and quality in output images. More in-depth analysis is in Sec. 5.4.1.

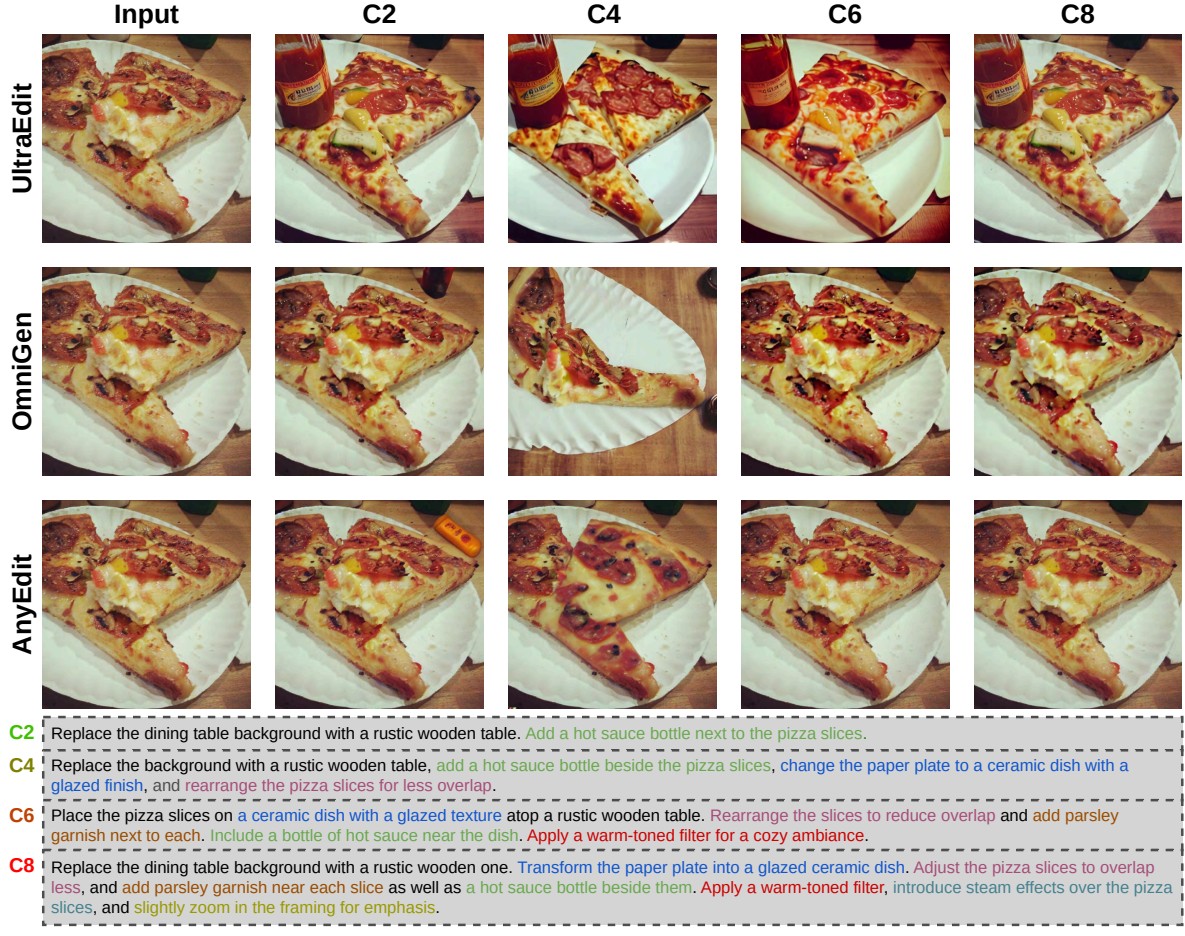

Figure 22: Qualitative results with sequential editing with Best-of-4 on real-life images. Identity Preservation and Perceptual Quality of sequential editing improve significantly with Best-of-4. Refer to Sec. 5.4.2 for more detailed discussions.

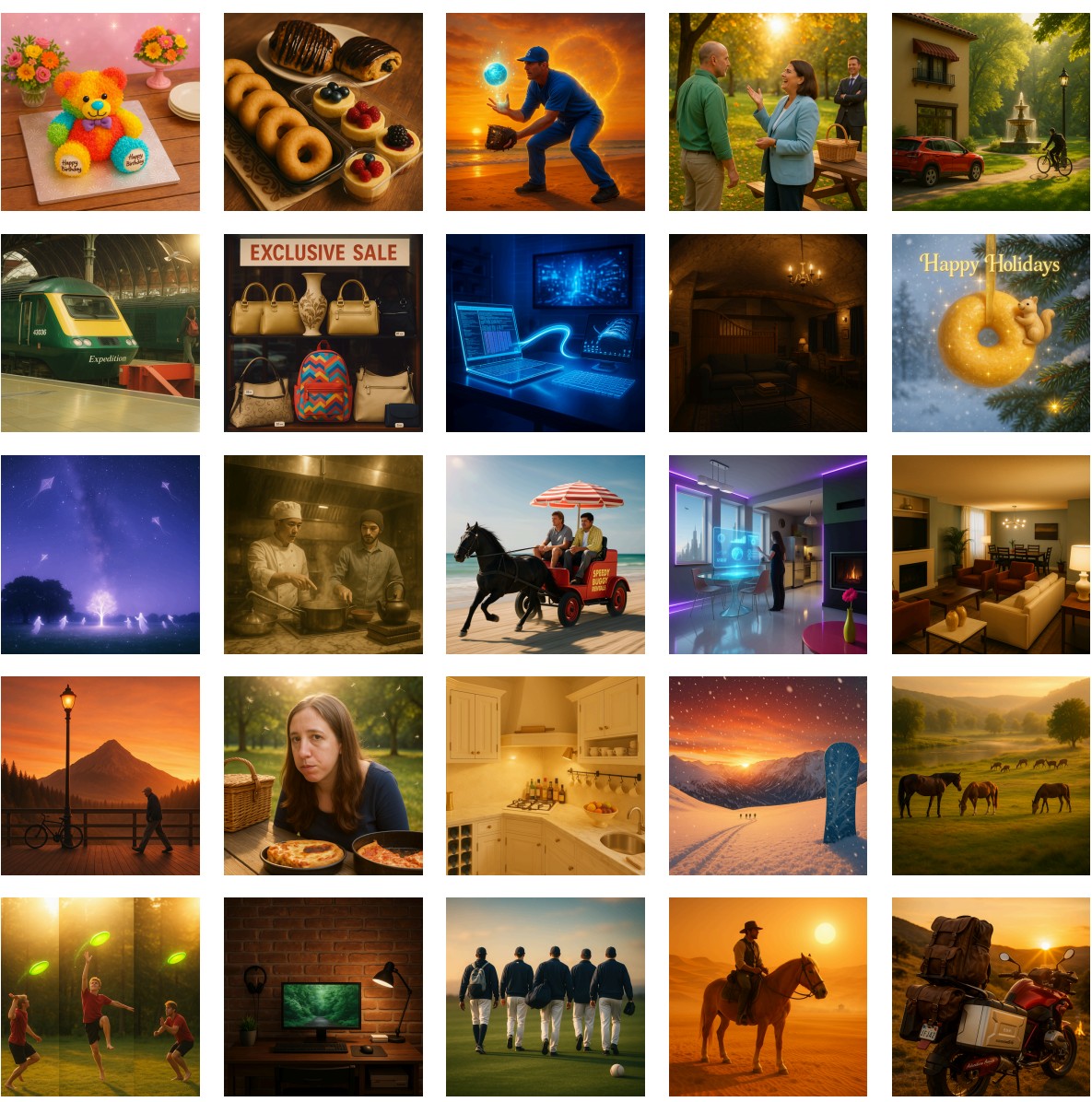

Figure 23: More real-life images edited with $C_8$ instructions by GPT-Image-1. Outputs from GPT-Image-1 severely lose the photorealism style. Refer to Sec. 5.3 for a detailed discussion.

# G   Assets and Licensing

All code used for evaluation and data generation is released at https://github.com/UCSC-VLAA/Complex-Edit. We also release the test set under the Creative Commons Attribution-NonCommercial 4.0 (CC BY-NC 4.0) license[1]. This release includes both real and synthetic input images, as well as all the artifacts produced across the three generation stages, including atomic instructions (both pre- and post-simplification), rationales, and the final composed instructions.

# H   Ethical concerns

We manually review all the generated content, including the synthetic input images generated by FLUX.1 (Labs, 2024) as well as the instructions and rationales from GPT-4o (Hurst et al., 2024), to ensure that there is no inappropriate generation.

---

[1]https://creativecommons.org/licenses/by-nc/4.0/

