# OpenReview forum: "\texttt{Complex-Edit}: CoT-Like Instruction Generation for Complexity-Controllable Image Editing Benchmark"
_TMLR — Accepted by TMLR_

### Review · Reviewer_69ut · 2025-11-23

**Summary Of Contributions:**

The paper introduces Complex-Edit, a benchmark for instruction-based image editing where instruction complexity is controlled by composing multiple atomic edit operations. A three-stage GPT‑4o-driven pipeline generates sequences of atomic instructions for each image, simplifies them, and compounds prefixes of length 1…8. The authors also design a VLM-based evaluation protocol that evaluates for instruction following, identity preservation, and perceptual quality. Using their benchmark, the authors evaluate multiple open- and closed-source models, analyze direct vs CoT-style sequential editing, and study Best-of‑N selection and the “curse of synthetic data” phenomenon.

**Strengths:**
- The paper presents several interesting findings.
- The 24 atomic instructions offer a good coverage of editing primitives.
- Splitting alignment metric into IF and IP seems like a good alternative to CLIP scoring, especially because it aligns more with human judgement.
- The paper provides a good analysis for their design choices: numeric scoring vs. token probabilities, CoT vs. no CoT, rubric vs. no rubric, and deterministic decoding vs. stochastic sampling.
- Evaluating three open-source models and three proprietary models on both real and synthetic images at different complexity levels provides a fairly comprehensive evaluation.

**Limitations:**
- Heavy dependence on GPT-4o as both instruction generator and evaluator, without a thorough bias analysis.
- Reliance on only EMU-Edit test images and relatively small scale.
- The paper does not clearly state whether the dataset (images, instructions, metadata) and evaluation code will be publicly released.

**Audience:**

Yes

**Audience Explanation:**

The paper offers a good evaluation of existing open- and closed-source image editing models across instructions of varying complexity and presents a well-designed evaluation protocol. The findings and strategies used would likely be of interest to many in the audience.

**Broader Impact Concerns:**

The dataset uses GPT-4o to automatically generate editing instructions (e.g., changing a person’s expression or body posture) and includes both synthetic and real-world images modified according to those instructions. Adding a Broader Impact section would help address the ethical considerations involved in generating and editing such data.

**Claims And Evidence:**

Yes

**Claims Explanation:**

I think the authors do a good job in substantiating the claims with thorough experiments.

**Requested Changes:**

I would request the following updates to the paper:
- Provide a clearer dataset description section and release plan.
- Add an ethics section addressing data licensing and misuse.

---

> ### Author Response · Authors · 2026-01-19
>
> We thank the reviewer for their constructive feedback and appreciation of our work. We are encouraged that the reviewer finds our Complex-Edit benchmark to offer meaningful findings and validates the coverage of our 24 atomic editing instructions. We also appreciate the positive recognition of our evaluation protocol—specifically the utility of splitting alignment into Instruction Following (IF) and Identity Preservation (IP)—as well as our comprehensive analysis of design choices and model evaluations. We address your specific concerns as follows:
>
> # Heavy dependence on GPT-4o for both data generation and evaluation
> We appreciate the reviewer raising this critical point regarding potential self-evaluation bias. To address this, we conducted an additional evaluation using Gemini-2.5-Pro, a VLM from a distinct model family, to cross-verify our findings. As detailed in Appendix D and Table 7, the evaluation results from Gemini-2.5-Pro align closely with those from GPT-4o. Specifically, the key trends—such as the widening performance gap between open-source and proprietary models and the relative rankings of models like GPT-Image-1—remain consistent across both evaluators. This consistency confirms that the benchmark’s conclusions are robust and not artifacts of model-family bias.
>
> # Image number too small
> We thank the reviewer for this comment and clarify that while we utilize the EMU-Edit test set as a foundation, the scale of our benchmark is defined by the editing tasks, not just the input images. Our pipeline generates 8 progressive complexity levels ($C_1$–$C_8$) for each of the 531 real and 531 synthetic images. Even focusing strictly on these complex instructions, this yields 8,496 unique evaluation samples (image-instruction pairs).This high density of samples per image is a deliberate design choice: it allows us to systematically measure performance degradation as a function of complexity—a dimension often overlooked in larger, 'flat' datasets. Empirically, this scale proves robust, yielding consistent and statistically significant insights across five diverse state-of-the-art models (both open-source and proprietary).
>
> # Missing release plan
> As mentioned in the abstract and Appendix F in our revised version, all code used for evaluation and data generation will be released upon publication, as well as the test set with all the generated assets, including rationales, instructions, and synthetic input images. For now, we present the code to be released in this [anonymous repo](https://anonymous.4open.science/r/Complex-Edit_re-5E8A).
>
> # Ethical concerns about misuse
> We appreciate the reviewer raising the issue of safety. We would like to clarify that Complex-Edit is an evaluation benchmark, not a generative model; therefore, it does not directly generate content for end-users. However, to ensure the benchmark itself is safe, we implemented a rigorous manual review process for all synthetic images, instructions, and rationales to ensure no inappropriate content is included, as detailed in revised Appendix G.

---

### Review · Reviewer_2iM2 · 2025-12-04

**Summary Of Contributions:**

### Summary of Contributions:
This paper introduces Complex-Edit, a large-scale benchmark designed to systematically evaluate instruction-based image editing models under varying levels of instruction complexity. The authors develop a three-stage GPT-4o-driven data generation pipeline (“Sequence Generation → Simplification → Instruction Compounding”) to create editing instructions with controllable complexity. They further propose an improved VLM-based auto-evaluation framework, incorporating refined metrics—Instruction Following (IF), Identity Preservation (IP), and Perceptual Quality (PQ)—along with detailed rubrics and variance control strategies to enhance alignment with human judgments. Using this benchmark, they conduct an extensive study across open-source and proprietary models, uncovering key phenomena such as the decline of identity preservation with complexity, the limited robustness of stronger models, and the “curse of synthetic data” where models trained with synthetic inputs produce increasingly artificial results under complex edits.

### Key Strengths:
- Comprehensive and systematic benchmark: Introduces the first dataset explicitly designed to evaluate complexity in instruction-based image editing.

- Robust evaluation methodology: Well-validated VLM-based auto-evaluation framework that correlates strongly with human judgment.

- Novel empirical insights: Reveals important trends—performance degradation under complexity, limits of test-time scaling, and synthetic data artifacts.

- Thorough experimental design: Includes multiple models (open-source and proprietary), realistic vs. synthetic data, and rigorous meta-evaluation of metrics.

### Key Weaknesses:

- Heavy dependence on proprietary GPT-4o for both dataset creation and evaluation, potentially limiting reproducibility and accessibility.

- Limited semantic diversity analysis: “Complexity” is defined mainly by the number of atomic edits rather than deeper compositional semantics.

- Potential circularity in evaluation: Using the same VLM family for generating and grading data could bias outcomes toward GPT-style reasoning.

- Qualitative analysis of realism drift is insightful but somewhat subjective, with limited quantitative grounding.

**Audience:**

Yes

**Audience Explanation:**

The paper’s findings would be of clear interest to a substantial portion of the TMLR audience.

### Rationale:

- The work directly addresses an emerging subfield at the intersection of vision-language modeling and generative editing, which overlaps with current research trends in multimodal learning, diffusion models, and instruction following.

- Its benchmarking and evaluation framework fill a recognized gap in how instruction-based image editing systems are assessed, offering practical tools and insights valuable to both academic and industry researchers.

- The discovery of phenomena such as the performance degradation with increasing edit complexity and the “curse of synthetic data” are conceptually significant and likely to stimulate further methodological and dataset research in multimodal generation and training data quality.

- Even readers outside image editing (e.g., those working on instruction tuning, evaluation metrics, or test-time scaling in multimodal models) would find the methodological innovations in auto-evaluation and meta-evaluation broadly relevant.

In short, the paper contributes both practical benchmarking resources and conceptual insights that align well with TMLR’s focus on rigorously analyzing and improving machine learning systems.

**Broader Impact Concerns:**

### Broader Impact Concerns:
The paper’s benchmark and evaluation pipeline raise several ethical considerations that merit explicit acknowledgment:

### Synthetic Data Feedback Loops:
The identified “curse of synthetic data” highlights a risk that repeated use of synthetic images in training and evaluation could erode visual realism and authenticity over time. The authors should briefly discuss safeguards against such self-reinforcing cycles.

### Automated Evaluation Bias:
Reliance on proprietary VLMs (GPT-4o, Gemini) for judging aesthetics and accuracy may embed opaque cultural or stylistic biases. A short note on the limits of automated evaluation and the need for periodic human calibration would improve transparency.

### Misuse Potential:
More capable instruction-based editing systems can be misused for deceptive or harmful image manipulation. Acknowledging this risk and referencing standard safety measures (e.g., watermarking, content filtering) would strengthen the ethical framing.

### Accessibility and Equity:
Since data generation and evaluation depend on paid APIs, researchers with limited resources may be excluded. Mentioning plans for partial data or prompt release could mitigate this concern.

Overall, these issues don’t undermine the technical work but warrant a Broader Impact Statement addressing synthetic-data amplification, evaluation bias, misuse prevention, and equitable access.

**Claims And Evidence:**

Yes

**Claims Explanation:**

Overall, yes — the paper’s claims are well supported by convincing and clearly presented evidence, though a few limitations remain.

### Strengths in Support and Evidence:

- The paper provides comprehensive quantitative evidence across multiple models (open-source and proprietary), two image domains (real vs. synthetic), and eight levels of instruction complexity. This breadth substantiates claims about general trends—especially the decline of identity preservation and the widening performance gap at higher complexity.

- The meta-evaluation study rigorously validates their VLM-based scoring framework against human judgments, lending strong credibility to their quantitative results.

- The reported findings are consistent across two evaluators (GPT-4o and Gemini-2.5-Pro), reinforcing robustness.

- Detailed qualitative visual comparisons (figures illustrating degradation, synthetic drift, and sequential editing artifacts) visually corroborate the numerical trends.

### Areas Less Strongly Supported:

- While the “curse of synthetic data” phenomenon is compelling, the evidence is mostly qualitative and interpretive rather than quantitatively isolated or controlled.

- The definition of instruction complexity (by number of atomic edits) is clear and systematic but may not capture all aspects of semantic or cognitive difficulty; this limits how far the conclusions can generalize.

- Because both data generation and evaluation depend heavily on GPT-4o, some bias or circularity may remain unmeasured.

### Verdict:
The paper’s core claims are accurate and convincingly backed by extensive experiments and analyses, with only minor caveats regarding the scope of “complexity” and the interpretive nature of certain findings (e.g., the synthetic-data effect).

**Requested Changes:**

## Critical Adjustments (Required for Acceptance)

### Clarify the Definition and Scope of “Complexity.”

- Issue: The paper defines instruction complexity solely as the number of merged atomic edits, which may not fully represent semantic or compositional difficulty.

- Adjustment: Provide justification or empirical evidence that this metric correlates with human-perceived complexity (e.g., via small-scale user validation or linguistic analysis).

- Rationale: Essential for ensuring the benchmark truly measures the intended construct and that results are interpretable beyond this operational definition.

### Address Potential Bias and Circularity in GPT-4o–Based Generation and Evaluation.

- Issue: GPT-4o is used both to generate the dataset and to evaluate models, raising the possibility of model-family bias.

- Adjustment: Include a discussion or experiment assessing how results differ when evaluated with a different VLM family (e.g., Gemini, Claude, or open-source BLIP/InternVL).

- Rationale: Critical to confirm that the benchmark’s conclusions are not artifacts of one model’s evaluation tendencies.

### Quantify the “Curse of Synthetic Data.”

- Issue: This claim is supported mainly by qualitative visuals and speculation about training data composition.

- Adjustment: Add quantitative measures (e.g., FID, realism classifiers, human realism ratings) or controlled experiments isolating the proportion of synthetic data.

- Rationale: Strengthens the credibility of this key empirical insight and distinguishes correlation from causation.

## Non-Critical Improvements (Would Strengthen the Work)

### Enhance Transparency and Reproducibility.

- Issue: Heavy reliance on proprietary systems (GPT-4o, Imagen3, SeedEdit) limits replicability.

- Adjustment: Clearly state which data and scripts can be released publicly and provide detailed pseudocode or prompts for the three-stage data-generation pipeline.

- Rationale: Necessary to ensure future researchers can replicate or extend the benchmark.

---

> ### Author Response · Authors · 2026-01-19
>
> We thank the reviewer for their thoughtful and detailed review. We are glad that the reviewer recognizes Complex-Edit as a comprehensive and systematic benchmark that effectively addresses the gap in evaluating instruction complexity. We also appreciate the positive feedback on our VLM-based evaluation methodology—specifically its robustness and strong correlation with human judgment—as well as the value of our novel empirical insights regarding performance degradation and the "curse of synthetic data." We address your specific concerns as follows:
>
> # Bias and circularity in evaluation
> We appreciate the reviewer raising this critical point regarding potential self-evaluation bias. To address this, we conducted an additional evaluation using Gemini-2.5-Pro, a VLM from a distinct model family, to cross-verify our findings. As detailed in Appendix D and Table 7, the evaluation results from Gemini-2.5-Pro align closely with those from GPT-4o. Specifically, the key trends—such as the widening performance gap between open-source and proprietary models and the relative rankings of models like GPT-Image-1—remain consistent across both evaluators. This consistency confirms that the benchmark’s conclusions are robust and not artifacts of model-family bias.
>
> # Clarify the scope of 'complexity'
> We appreciate the reviewer suggesting that we clarify the definition and scope of "complexity." In the revised Sec. 1, we explicitly state that our benchmark targets Semantic Complexity (specifically Compositional and Contextual Complexity), as opposed to abstract reasoning or knowledge-intensive tasks.
>
> Within this scope, we justify using the number of atomic operations as the definition of complexity because it directly dictates the compositional load of the task: increasing the number of operations linearly scales the linguistic density and compels the model to resolve increasingly intricate attribute binding and disentanglement challenges.
>
> To further contextualize this definition, we have introduced a comprehensive 'Hierarchy of Editing Complexity Types' in the revised Appendix A, which distinguishes our focus from simple editing tasks and inferential-complex editing tasks.
>
> # Quantitative analysis on the "curse of synthetic data"
> We appreciate the opportunity to elaborate on the 'curse of synthetic data.' As noted in our submission, we observed that models heavily trained on synthetic data (such as UltraEdit) or highly capable proprietary models (like GPT-Image-1) tend to produce outputs that look increasingly synthetic under complex instructions.
>
> To support these qualitative observations with hard data, we are currently finalizing a rigorous set of experiments. We plan to employ a discriminator-based auto-evaluation or a human evaluation to assess perceptual realism. We anticipate sharing these results later this week.
>
> # Transparency, reproducibility, and accessibility
> As mentioned in the abstract and Appendix F in our revised version, all code used for evaluation and data generation will be released upon publication, as well as the test set with all the generated assets, including rationales, instructions, and synthetic input images. For now, we present the code to be released in this [anonymous repo](https://anonymous.4open.science/r/Complex-Edit_re-5E8A).
>
> # Synthetic data feedback loops
> Thank you for highlighting this critical issue. We have revised Section 5.3 to explicitly address the risks of self-reinforcing synthetic cycles. The updated text emphasizes the necessity of safeguards, including balancing synthetic data with high-quality real-world examples and implementing rigorous human-in-the-loop verification steps.
>
> # Ethical concerns and misuse potential
> We appreciate the reviewer raising the issue of safety. We would like to clarify that Complex-Edit is an evaluation benchmark, not a generative model; therefore, it does not directly generate content for end-users. However, to ensure the benchmark itself is safe, we implemented a rigorous manual review process for all synthetic images, instructions, and rationales to ensure no inappropriate content is included, as detailed in the revised Appendix G.

---

> > ### Author Response · Authors · 2026-01-27
> >
> > In the revised Appendix E, we have included human evaluation results based on a 300-image sample to further substantiate our claims regarding the "Curse of Synthetic Data." We deliberately prioritize human assessment over VLM-based automatic metrics due to the fantastical or physically impossible elements in our image-editing instructions. Evaluating the "realism" of such content requires more than just visual accuracy; it necessitates counterfactual reasoning, which remains a uniquely human cognitive capability.

---

### Review · Reviewer_nkmc · 2026-01-04

**Summary Of Contributions:**

This paper proposes a  a large-scale benchmark called Complex-Edit, designed for evaluating instruction-based image editing models across instructions of varying complexity. The benchmark is constructed using Chain-of-Edit data generation pipeline, which constructs complex editing instructions by compounding atomic editing operations. GPT-4o is used to automatically collect a diverse set of editing
instructions at scale. The paper also proposes a VLM-based evaluation framework based on three types of metrics: Instruction Following, Identity Preservation, and Perceptual Quality. In addition to the benchmark construction, experiments on five image editing models reveal several valuable empirical findings, such as the degradation under increasing complexity, the failure of CoT-like sequential editing, and the “curse of synthetic data”.

Strengths:
- Paper is well-written and easy to follow
- Data generation pipeline is clearly defined, structured (24 operations across 9 categories are defined), is scalable and diverse
- Evaluation metric is divided into three meaningful categories (Instruction Following, Identity Preservation and Perceptual Quality).
- Correlation with human judgment is investigated, which is important for the task of image editing, where the aim is to basically increase human satisfaction with the result.
- Good ablation on metrics (numeric scoring vs token-probability scoring, effect of rubrics)
- Experimental results provide interesting insights, such as effect of CoT, instruction input on VLM evaluation, and missing correlation between model strength and improved image editing results
- 5 modern image editing models are considered, including open-sourced and proprietary ones.

Weaknesses:
- For generating synthetic images, only FLUX.1 is used. It is interesting to see if the choice of the model affects image editing performance, as there is evidence that most image generative models produce images with certain distinct characteristics ([1], [2]) which possibly can impact on how well image editing models can process these images
- As far as I understand, in the Best-of-N setting, the metric that is used to select the best candidate is the same as the final metric used to assess the editing quality (with the same VLM). I am not sure if this is the standard thing to do, but in my mind this creates a situation when a final image might become biased towards metric computed by a certain VLM. Does it make sense to do so?
- In fig.5. I personally do not understand why the last image in the second row has the highest identity preservation score. Doesn't a second image from the right follow the original image better?
- "Curse of Synthetic data" is presented only in the qualitative setting. It would be nice to see some metrics, e.g. CLIP alignment score with the prompts of a form "an image with synthetic aesthetic" and  "an image with realistic aesthetic" or asking some VLM to assess realism of an input image. This is an important finding, in my opinion, hence it's desirable to support it with quantitative evidence.
- It would be interesting to see analysis of failure mods of image editing models. The presented benchmark consists of 24 operations divided by 9 categories, and it would be interesting to see if there is a pattern of which categories/operations fail first in which image editing models. I can imagine that some edits (for example, involving complex/small objects) resulting in more failures than others (e.g. style manipulations). Also, it would help reveal biases of image editing models towards certain types of edits. Next, more complex prompts are constructed as combinations of simple editing prompts: is there evidence that some combinations are more challenging for the models to handle than others (e.g. background + lighting, object + color, etc)?  Also, it would be interesting to see reasons why different models fail on benchmark tasks. E.g. is failure defined by hallucinating new objects, object omission or identity drift? This might shed a light on how to improve these models and bridge the gap between open-source models and proprietary models.
- Overall metric is now computed as a simple average of three components (Instruction Following, Identity Preservation and Perceptual Quality). Is it the best way to compute a final metric? Intuitively, I can imagine that some of these components (e.g. Perceptual Quality) mean more than others (e.g. Identity Preservation). Would other weighting strategies improve correlation with human judgement?


[1] Guangyu Nie, Changhoon Kim, Yezhou Yang, Yi Ren: Attributing Image Generative Models using Latent Fingerprints. ICML 2023: 26150-26165
[2] Hae Jin Song, Mahyar Khayatkhoei, Wael AbdAlmageed: ManiFPT: Defining and Analyzing Fingerprints of Generative Models. CVPR 2024: 10971-10981

**Audience:**

Yes

**Audience Explanation:**

The construction of a well-defined, complexity-controllable benchmark with a clear evaluation framework for assessing the quality of image editing models is an important and highly relevant topic

**Broader Impact Concerns:**

-

**Claims And Evidence:**

Yes

**Claims Explanation:**

The paper is clearly written and coherent in general

**Requested Changes:**

I would request the following:
- I would like the authors to answer my question about Best-of-N setting
- Assessing "Curse of Synthetic data" using some relevant quantitative measure
- Analysis of failure mods of image editing models (at least assessment of individual complexities of atomic edits: are there edit types that are more challenging than others in general? Can we treat them equally?)

---

> ### Author Response · Authors · 2026-01-19
>
> We thank the reviewer for their detailed and constructive comments. We are encouraged that the reviewer finds our paper well-written and recognizes the proposed Complex-Edit benchmark as clearly defined, scalable, and diverse. We also appreciate the positive feedback on our VLM-based evaluation framework—specifically its meaningful categorization and correlation with human judgment—as well as the value of our empirical findings regarding the limitations of CoT and the "curse of synthetic data." We address your specific concerns point-by-point below:
>
> # Alternative source for synthetic input images
> We thank the reviewer for this insightful suggestion and agree that distinct generative artifacts can influence editing performance. We prioritized FLUX.1 specifically for its high visual fidelity and strong instruction adherence. Unfortunately, generating a parallel dataset with an additional model would nearly double our input generation costs, which exceeds our current computational resources.
>
> Despite this limitation, we believe our work offers substantial contributions to the field. We have presented extensive findings regarding the performance gap between open and closed-source models, the detrimental effects of instruction complexity on content retention, and the counterintuitive degradation observed in step-by-step editing approaches. Additionally, we have **included the new Appendix A** in the revision, "The Hierarchy of Editing Complexity Types," to further formalize our definition of instruction complexity.
>
> # Justification for the Best-of-N setting
> First, there are such practices in published literature to employ the same model or metric as both the verifier during test-time scaling and the evaluator for final assessment. For instance, Ma et al.[1] utilize Aesthetic Score, CLIPScore, and ImageReward for both the search process and the final evaluation on DrawBench.
>
> Furthermore, as detailed in the revised Appendix D, we provide evaluation results using Gemini-2.5-Pro under the direct editing setting. These results align closely with those from the GPT-4o evaluator, demonstrating consistency across two distinct model families. This substantiates the assumption that both models share similar evaluation preferences—likely due to their strong alignment with human preference—and suggests that evaluating GPT-4o-selected Best-of-N outputs with Gemini-2.5-Pro would yield consistent conclusions.
>
> # Clarification about Fig. 5 and IP evaluation
> According to GPT-4o's rationale for this evaluation, the main reason for the point deduction for the second rightmost image is the wrongly generated vehicle on the left side.
> Also, although the second rightmost image seems to be similar to the input image, the evaluation of Identity Preservation is to focus on whether the elements that are not supposed be changed remain unchanged instead of whether the overall output image is close to the original image. This is the reason why traditional metrics, such as CLIP scores, that are solely based on image similarity without taking editing instructions into account is not suitable for image editing tasks.
>
> # Additional quantitative analysis on the "curse of synthetic data" and atomic operation types
> We are currently finalizing the relevant experiments to address this. For assessing the "curse of synthetic data", we plan to employ a discriminator-based auto-evaluation or human evaluation. Additionally, we are conducting the requested fine-grained failure analysis to identify specific breakdown patterns across the 9 operation categories. We will share these quantitative results and detailed conclusions later this week.
>
> # Alternative way of computing overall metric
> As discussed in Sec. 4.1.5, we experimented with two approaches to calculate the overall score, that are the geometric mean and the arithmetic mean, and we found that the latter yields a significantly higher correlation with human evaluation (0.23324 vs 0.386). We didn't assign different weights for different components out of the motivation to keep the metric calculation straightforward, and the ad hoc designs minimal.
>
> [1] Ma et al., Inference-time scaling for diffusion models beyond scaling denoising steps. In CVPR 2025.

---

> > ### Author Response · Authors · 2026-01-27
> >
> > We just provided the brief human evaluation results based on 300 images regarding the "Curse of Synthetic Data" in the revised Appendix E, further supporting our existing claim. We use only human evaluation rather than VLM-based automatic assessment because image-editing prompts often include fantastical or physically impossible elements (e.g., science-fiction or magical content). Judging the “realism” of such images requires not just precise visual perception but also counterfactual reasoning, such as inferring how a unicorn should plausibly look if unicorns existed.

---

### Decision · Action_Editor_4Lme · 2026-02-12

**Recommendation:** Accept as is

**Audience:**

Yes

**Audience Explanation:**

This paper addresses problems at the intersection of vision-language modeling and generative editing, with relevance to multimodal learning, diffusion models, and instruction following. Thus, it would be relevant to a broad TMLR audience. The paper provides both practical benchmarking resources and insights. The findings would interest readers in multimodal generation as well as those in other areas where methodological insights such as auto-evaluation may be relevant.

**Claims And Evidence:**

Yes

**Claims Explanation:**

All reviewers recommended accepting this paper. The paper is well-written, with a clearly defined data generation pipeline and evaluation metrics. It presents comprehensive experiments with diverse coverage of image editing models and atomic instructions for editing. The experiments include extensive ablation studies and provide interesting insights. Correlation between automatic evaluation and human judgment has been demonstrated. The main remaining concern is the exclusive use of FLUX.1 for synthetic data generation, but it is relatively minor.